# Worldwide dynamic biogeography of zoonotic and anthroponotic dengue

**Alisa Aliaga-Samanez**[1]*, **Marina Cobos-Mayo**[1], **Raimundo Real**[1,2], **Marina Segura**[3], **David Romero**[1,4], **Julia E. Fa**[5,6], **Jesús Olivero**[1,2]

**1** Grupo de Biogeografía, Diversidad y Conservación, Departamento de Biología Animal, Facultad de Ciencias, Universidad de Málaga, Málaga, Spain, **2** Instituto IBYDA, Centro de Experimentación Grice-Hutchinson, Málaga, Spain, **3** Centro de Vacunación Internacional de Málaga, Ministerio de Sanidad, Consumo y Bienestar Social, Málaga, Spain, **4** Laboratorio de Desarrollo Sustentable y Gestión Ambiental del Territorio, Facultad de Ciencias, Universidad de la República, Montevideo, Uruguay, **5** Division of Biology and Conservation Ecology, Manchester Metropolitan University, Manchester, United Kingdom, **6** Center for International Forestry Research (CIFOR), CIFOR Headquarters, Bogor, Indonesia

* alisaliaga@uma.es

## Abstract

Dengue is a viral disease transmitted by mosquitoes. The rapid spread of dengue could lead to a global pandemic, and so the geographical extent of this spread needs to be assessed and predicted. There are also reasons to suggest that transmission of dengue from non-human primates in tropical forest cycles is being underestimated. We investigate the fine-scale geographic changes in transmission risk since the late 20[th] century, and take into account for the first time the potential role that primate biogeography and sylvatic vectors play in increasing the disease transmission risk. We apply a biogeographic framework to the most recent global dataset of dengue cases. Temporally stratified models describing favorable areas for vector presence and for disease transmission are combined. Our models were validated for predictive capacity, and point to a significant broadening of vector presence in tropical and non-tropical areas globally. We show that dengue transmission is likely to spread to affected areas in China, Papua New Guinea, Australia, USA, Colombia, Venezuela, Madagascar, as well as to cities in Europe and Japan. These models also suggest that dengue transmission is likely to spread to regions where there are presently no or very few reports of occurrence. According to our results, sylvatic dengue cycles account for a small percentage of the global extent of the human case record, but could be increasing in relevance in Asia, Africa, and South America. The spatial distribution of factors favoring transmission risk in different regions of the world allows for distinct management strategies to be prepared.

**Data Availability Statement:** The data with the compendium of dengue cases from 2013 to 2017 that support the findings of this study are available

## Author summary

The rate of disease emergence is increasing globally, and many long-existing diseases are extending their distribution ranges. This is the case for dengue, a global pandemic whose mosquito vectors are currently occupying ever-increasing numbers of regions worldwide. We updated the most complete global dataset of dengue cases available, and addressed the

in Dryad Digital Repository at https://doi.org/10.5061/dryad.9w0vt4bfv.

**Funding:** This study was supported by the Project CGL2016-76747-R, of the Spanish Ministry of Economy, Industry and Competitiveness and the European Regional Development Fund. AA-S was supported by the FPU16/06710 grant of the Spanish Ministry of Education. JEF was funded by USAID as part of the Bushmeat Research Initiative of the CGIAR research program on Forests, Trees and Agroforestry. The funders had no role in study design, data collection and analysis, decision to publish, or preparation of the manuscript.

**Competing interests:** The authors have declared that no competing interests exist.

fine-scale analysis of the geographic changes experienced in dengue-transmission risk since the late 20[th] century. Our approach is the first to take into account the potential role of primates and sylvatic vectors in increasing the disease transmission risk in tropical forests. We built models that describe the favorable areas for vector presence and for disease occurrence, and combined them in order to obtain a novel model for predicting transmission risk. We show that dengue transmission is likely to spread to affected areas in Asia, Africa, North and South America, and Oceania, and to regions with presently no or very few cases, including cities in Europe and Japan. The global contribution of sylvatic dengue cycles is small but meaningful. Our methodological approach can differentiate the factors favoring risk in different world regions, thus allowing for management strategies to be prepared specifically for each of these regions.

## Introduction

Dengue is a viral disease caused by the dengue virus, a group of four Flaviviridae serotypes [1]. The pathogen is principally transmitted by female mosquitoes of the genus *Aedes* to humans. In most cases, the pathogen causes mild illness, but is also known to cause flu-like symptoms, occasionally producing severe complications that are fatal [2]. More than 14,000 annual deaths are reported [3]. Dengue infections occur mainly in the Asian, African, and American tropics, but are being reported in many regions worldwide. The rapid spread of dengue is considered to represent a global pandemic threat [4].

The World Health Organization (WHO) reports that annual dengue cases have increased from approximately 500,000 in 2000 to approximately 4.2 million in 2019 [2]. This is considered an underestimation, however. Some authors have calculated that in 2013 between 58.4 million [3] and 96 million [5] yearly cases may have occurred worldwide, with many cases remaining unreported [6] and other cases being mistaken for similar pathologies [7, 8]. This confusion presents serious challenges for assessing the scale and geographic extent of the risk of disease transmission. However, distribution modelling has been used to map the global risk [5, 9].

Distribution models are useful not only for locating risk hotspots [10, 11] but also can be employed to inform prevention and mitigation strategies [12, 13] such as vector control measures, large-scale vaccination programmes, and traveller health-care advice. The first global dengue model produced at a high resolution described the geographic distribution of the risk of dengue transmission for the period 1960–2010 [5]. According to this model, as many as 390 million dengue infections in 128 countries were predicted [14], in contrast to the 4.2 million cases recorded in 2019 [2]. Annual records of dengue transmission [15] up to 2015 have provided data for the generation of risk models [9], which have also considered the environmental suitability for the arbovirus vectors *Aedes aegypti* and *Ae. albopictus* as covariables [16]. The integration of known and potential reservoir species in disease distribution models has already proved invaluable in pathogeography [17, 18] whereas the design of disease models that reflect complex interactions has benefited from biogeographical concepts and tools [18–20]. Thus, determining the distribution of infectious diseases needs to take into account the patterns of distribution of reservoirs and/or vectors [21, 22] and the ecology of the pathogen itself [23], which involves consideration of the environment as well as the human-geography context.

Dengue is principally an indirectly transmitted anthroponosis [24], humans being the main hosts and *Aedes* mosquitoes the main vectors. However, there are zoonotic "sylvatic" cycles in Africa and Asia where non-human primates are asymptomatically infected by the dengue

virus, which is efficiently transmitted by the mosquito fauna in these regions [25–27]. There is evidence to suggest that the virus originated in monkeys, and that every one of its four serotypes was independently transmitted to humans in Africa and Asia [27, 28]. Transmission to humans from other primates appears to be infrequent, but they do occur. The scarcity of records might be a result of inadequate characterization of human exposure to sylvatic viruses [28]. In Africa and in Asia, there is high potential for the re-emergence of sylvatic dengue in the human transmission cycle as a result of deforestation, climate change, and vector geographic expansion [28]. While the existence of sylvatic dengue cycles has not been demonstrated in the Neotropic realms, there are reasons to foresee this possibility. This is because enzootic cycles based on American primates are involved in the diversification and transmission to humans of the yellow-fever virus, which shares vectors with the dengue virus [29–31].

Sylvatic cycles in South America appear to shape the evolutionary dynamics of recent yellow-fever-virus lineages, and are involved in the current re-emergence of this virus in Brazil [32]. In this country, dengue-virus-RNA has been found in sylvatic mosquitoes that are vectors of the yellow-fever virus [33]. In addition, dengue-virus infections in humans may have occurred in Bolivia in the absence of *Ae. aegypti* and *Ae. albopictus* [34], and different mammalian taxa have been infected with dengue-virus in French Guiana [35]. In light of these data, the possible presence of sylvatic dengue in the American continent is likely [28, 36].

Available models defining the global risk of dengue transmission have not included a zoonotic component. Although likely to be negligible at a global scale, sylvatic-dengue cycles may increase local or regional risk amplification, or represent a threat of dengue re-emergence or diversification. In addition, the global risk of transmission to humans is increasing as a consequence of globalization [37]. The pathogen is easily transported by travelers [38], and there is a rapid expansion of the main vectors [39, 40], which also evolve as they spread [41]. This means that modelling approaches involving temporal stratification are required in the production of dengue-risk maps, as well as a multidisciplinary approach as proposed by the international *One Health* initiative [42], to consider a multifactorial dynamic. Hence, evaluations of the current risk should take into account a combination of the inertia of past times, the advent of new factors capable of changing previous expectations, and the zoonotic dimension. Here, we adopt a multitemporal and multifactorial pathogeographic approach to analyzing the risk of dengue transmission to humans. We produced a risk model for the early 21st century, using available information on dengue cases up to 2019. We achieve this under the assumption that transmission between humans is (1) limited by the vector presence, (2) constrained by environmental conditions favoring vectorial capacity [43], (3) could be locally or regionally favored by the occurrence of enzootic cycles in the tropical forests, and (4) are experiencing an inter- and intra-continental spread that is subject to the growth of both virus' and vectors' ranges. Our aim is to contribute to the international health system with reliable forecasts on areas where dengue transmission between humans could increase in the near future, and with quantification and mapping of the contribution of sylvatic cycles.

## Methods

### Study area and time period

All spatially explicit data (i.e., dengue case records, mosquito occurrences, primate ranges, environmental variables) were projected onto a worldwide grid composed of 18,874 hexagonal units of 7,774 km$^2$, built using Discrete Global Grids for R [44]. In this way, we prevented autocorrelation that could result from spatial dependence among very close occurrences [45].

As the dengue spatial trends are dynamic, the temporal extent for analysis purposes was divided into three periods: 1970–2000 ("the late 20th century"), 2001–2017 ("the early 21st

century"), and 2018–2019. Pre-1970 records were not considered so as to limit our conclusions to a contemporary setting. Some milestones regarding the fight against arboviral diseases occurred circa 1970, such as the re-infestation of Latin America with *Ae. aegypti*, the main vector of the dengue virus, after 50 years of eradication efforts [46]. The use of DDT was suspended in the late 1960s in several countries of the Americas due to resistance [47]. Although the limit between periods at the turn of the century is arbitrary, it reflects distributional changes in the ranges of the two urban *Aedes* vectors as well as the increase of case reports in sizeable regions all over the world—being, for example, previous to any contemporary record of autochtonous *Aedes*-born disease transmission in Europe and Japan. The bases of the current globalization of international movements were established in the last decade of the 20th century (e.g., with the fall of the Iron Curtain, the advent of the Internet, and the start of low-cost flights); and their full potential was reached just after the start of the 21st century (e.g., with the opening of international borders, the widespread access to the Internet and to cell phones, and the online travel booking generalization). In fact, from 1970 to 2000, global exports nearly doubled to approximately a quarter of global GDP [48]. The time stratification also provides further opportunities for model validation. Thus, predictions afforded by the late 20th-century models were validated using early 21st-century datasets, and validations of the early 21st-century-model predictions were addressed with post-2017 records. In addition, by performing separate analyses for both centuries, we were able to integrate, in our 21st-century models, social and ecological descriptors that are only available for the last two decades.

## Methodological framework

The ultimate objective of our analyses was to build a map that quantified the current level of dengue transmission risk worldwide. This map results from the combination of both a model describing favorable areas for the presence of vectors, and a model describing favorable areas for the occurrence of disease cases. These models were based on the predictive power of macro-environmental and spatial variables that included climate, topo-hydrography, vegetation, human activity, spatial autocorrelation, and potential for enzootic transmission. We first produced models focused on the late 20th century, which were later updated for the early 21st century through a procedure that involved reparameterization and the addition of variables representing changes in the distribution of the modelled item (i.e., of vector presences and dengue cases). The rationale for this addition is that, when animal and pathogen species spread, their distribution at a given moment is influenced by (1) the inertia of previous situations (i.e., temporal autocorrelation), here represented by the late 20th-century model, and (2) by a multifactorial set of new drivers potentially favoring the spread (i.e., spatial autocorrelation, environmental and socio-economic factors). A schematic description of our methodological framework is represented in Fig 1. The outputs of these models were expressed as favorability values (*F*, ranging 0–1) that represent the degree to which environmental conditions, at a particular spatial unit, favor the occurrence of a given event. Thus, favorability is equivalent to a degree of membership in the fuzzy set of environmentally favorable units [49], so that models based on favorability can be compared and combined through the implementation of fuzzy-sets theory tools [50]. *F* was calculated according to the Favorability Function [49, 51], defined by the following formula:

$$F = \frac{P}{1-P} \Big/ \left( \frac{n_1}{n_0} + \frac{P}{1-P} \right)$$

where *P* is the probability of occurrence of the event in question, $n_1$ is the number of recorded occurrences, and $n_0$ is the number of units in which occurrences have not been recorded. *P*

## Vector models

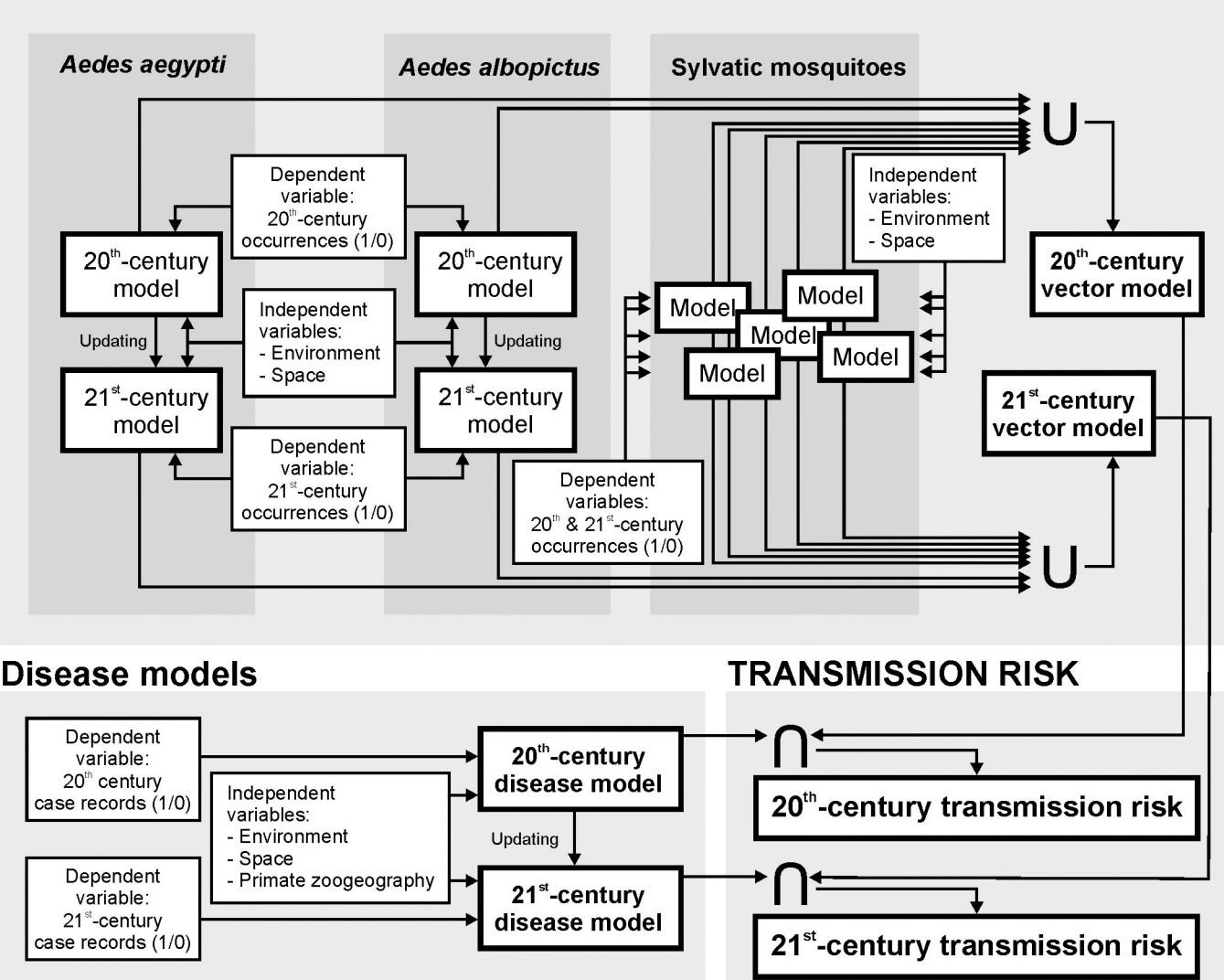

**Fig 1. Methodological framework for dengue transmission risk modelling. Vector models** result from combining, through the fuzzy union (U), favorable areas for the presence of urban and sylvatic vectors, thus denoting that the presence of one vector species already implies some potential for disease transmission to humans if the pathogen is present. For a given time period and vector species, a vector model is built using mosquito occurrences as dependent variables, and spatial/environmental descriptors as independent predictor variables. **Disease models** describe the areas favorable to the occurrence of dengue cases, using the presence/absence of dengue-case records as dependent variables, and spatial/environmental/zoogeographic descriptors as independent predictor variables. A temporal stratification differentiating between the late 20th century and the early 21st century was applied when the modelled item was subject to a temporally changing dynamic, i.e. to the global distribution of *Ae. aegypti*, *Ae. albopictus*, and dengue cases. 20th-century models were updated by complementing their equations with new variables capable of accounting for the observed changes of distribution. Finally, **transmission-risk models** quantify the level of dengue-transmission risk, according to the fuzzy intersection (∩) between vector and disease models. The intersection reflects that, for dengue to be transmitted in a given location, two elements, acting as limiting factors, must coincide in the area: 1) suitable environmental conditions for disease cases to occur; and 2) suitable conditions for the presence of vectors. Complete methodological descriptions are provided in the main text.

values were calculated through forward-backward stepwise logistic regression (using IBM SPSS Statistics 23), in which predictor variables were selected according to Rao's score tests [52], and derived from the formula:

$$P = \frac{e^y}{1 + e^y}$$

where *e* is the basis of Napierian logarithms, and *y* is the "logit equation", i.e. a linear combination of the predictor variables selected. We used iterative log-likelihood maximization for *y*-coefficient parameterization using a gradient ascent machine learning algorithm, and Wald tests [53] for evaluating the contribution of every variable in a model. The forward-backward stepwise approach prevents redundancy between variables in a model, as variable removal along the stepwise variable selection is allowed. Nevertheless, we strengthened prevention against excessive multicollinearity by preventing variables with Spearman correlation coefficients >0.8 from coinciding in the same model [18]. In case this happened, the least significant variable was deleted and the model was trained again. Benjamini and Hochberg's [54] procedure for calculating the False Discovery Rate (FDR) was followed to minimise Type I errors that could occur from the consideration of a large number of variables.

## Vector models

We built a global database of dengue vectors on the grid of 7,774-km$^2$ hexagonal units, through the projection of georeferenced records into hexagons, using ArcGIS 10.3. Records on mosquito species involved in the urban cycle, i.e., *Ae. aegypti* and *Ae. albopictus*, for the period 1970–2014, were taken from "The global compendium of *Aedes aegypti* and *Ae. albopictus* occurrence" [55] (S1 Table). Later occurrences were retrieved from the expert-validated citizen-science platform Mosquito Alert (http://www.mosquitoalert.com/) and from VectorBase (https://www.vectorbase.org/).

Available records on sylvatic vectors [28, 56–58] (mosquito species *Ae. polynesiensis*, *Ae. luteocephalus*, *Ae. africanus*, *Ae. niveus and Ae. vittatus*) were obtained from the literature (S2 Table), Vectormap (vectormap.si.edu), and Gbif (https://gbif.org) (S1 Table).

A worldwide favorability model was built for each mosquito species, using presence/absence of occurrence records at each hexagon as dependent variables, and environmental (i.e., climate, topo-hydrography, vegetation, and human-activity) descriptors as independent predictor variables (S3 Table). Only variables that can be considered reasonably stable in the short term were used at this stage of the analysis, due to the scarcity of high-resolution data for the late 20$^{th}$ century and the changing nature of environments during the study period. Thus, climate was represented by average values for the period 1979–2013 [59], vegetation was described using terrestrial ecoregions [60], and the human factor was represented by the distance to populated settlements [61] (thus avoiding having to change parameters such as population density, land use, and infrastructure). In addition to environmental variables, we used a trend surface approach [62] to account for purely spatial factors linked to contagious evolutionary and ecological processes preventing or promoting distribution shifts [62, 63]. The spatial factor could distinguish areas with similar environmental conditions but different probabilities of being reached by a spreading species. This could happen because of spatial autocorrelation (i.e., the species could have nearby populations in some cases); or it could be a result of recent introductions or reintroductions. On every continent that the species occur, we developed a favorability model based on purely spatial descriptors [45] (i.e., 1$^{st}$ to 3$^{rd}$-degree polynomial combinations of latitude and longitude). Then we added the resulting spatial-model outputs to the set of environmental variables.

In the case of urban-cycle mosquitoes, we generated favorability models based on late 20$^{th}$-century occurrence records, subsequently updated for the early 21$^{st}$ century. We updated it by developing a model based on early 21$^{st}$-century occurrence records. This model was completed in two blocks: (1) forcing the entry, as predictor variable, of the late 20$^{th}$-century-model logit equation; and (2) performing a later stepwise selection in which only variables with the

potential to account for changes with respect to the late 20th-century model were selected. This two-block variable selection was implemented using IBM SPSS Statistics 23.

Models for sylvatic-cycle mosquito species were run without temporal considerations, using the above-mentioned set of predictor variables. This is justified by the scarcity of occurrence records available for these species and by the assumption that their ranges have not changed substantially during the study period, which is based on a comparison of descriptions in the historical literature [58, 64, 65].

A vector favorability model for the late 20th century was produced using a combination of all individual vector models, including those for the sylvatic vectors and those for the urban vectors. We used the fuzzy union [66] for this purpose, which consisted of selecting, for every hexagon, the highest favorability value among those obtained in an individual model. The rationale for this criterion was that, if the pathogen is present in the area, the mere presence of one vector species already implies some potential for disease transmission to humans. Similarly, we created a vector model for the early 21st century.

## Disease model

The global record of dengue cases was projected onto 7,774-km$^2$ hexagons using ArcGIS 10.3. Georeferenced cases for the period 1970–2017 were obtained from the Messina et al.'s database [9], and considered only if they matched with the following criteria: (1) they were referred to precise locations, or (2) they were referred to centroids of polygons whose extensions were lower than or similar to the size of the hexagons (S1 Table). These data were completed with reports from Promedmail.org, using "DENGUE" as the keyword and limiting the search to the period 2013–2019, and with data provided by the epidemiological bulletins and weekly epidemiological surveillance of the Ministries of Health from Brazil, Costa Rica, Colombia, Ecuador, United States of America, Philippines, Honduras, Malaysia, Mexico, Myanmar, Palau, Puerto Rico, Samoa, Sri Lanka, and Thailand. In addition, we carried out searches in reports published by the European Centre for Disease Prevention and Control (ECDC): Communicable disease threats to public health in the European Union—Annual Epidemiological Report; and by the WHO: Dengue Situation Updates. Case reports for Africa were complemented with the Weekly Bulletin on Outbreaks and Other Emergencies (WHO, African Region), and publications available since 2017. Further information was obtained from the WHO and the Pan American Health Organization (PAHO) websites, and from the Global Infectious Disease and Epidemiology Online Network (GIDEON) [67].

We built a worldwide disease favorability model using presence/absence of case records at each hexagon as the dependent variable, and spatial/environmental descriptors as independent predictor variables (S3 Table). We used a similar methodological procedure as for vector species, including the performance of a late 20th-century model and its later update based on the early 21st century. The only difference with respect to the vector models was the inclusion of zoogeographical information in the set of predictor variables. This information defined the types of distribution ranges (i.e., chorotypes) of non-human primates, i.e., the most probable dengue reservoirs in the sylvatic cycles. A chorotype is a particular distribution pattern shared by a group of species, and may result from ecological and/or historical causes [68]. When knowledge of the reservoir-species complex is imprecise, the consideration of variables defining chorotypes shared by potential reservoirs helps to improve risk models referred to the distribution of zoonotic disease transmission [18, 23]. These primate-chorotype variables were defined in six steps:

1. Range maps of the African, Asian, and American primate species were obtained from the the IUCN [69], and were projected onto the grid of 7,774-km$^2$ hexagons to produce a

presence/absence matrix. The surface area of these units approximates the resolution below which extent-of-occurrence maps provided by the International Union for Conservation of Nature (IUCN) should not be employed for the characterization of macroecological patterns [70].

2. Chorotype analyses for each continent were addressed separately.

3. Primate ranges were classified hierarchically according to the Baroni-Urbani & Buser's similarity index [71], using the unweighted pair-group method using arithmetic averages (UPGMA) [72, 73].

4. All clusters in the resulting classification dendrogram were assessed for statistical significance using the method proposed by Olivero and colleagues [19], which uses RMacoqui 1.0 software (http://rmacoqui.r-forge.r-project.org/). Groups of distributions that were significantly clustered were considered chorotypes.

5. For each chorotype, a predictor variable was defined using the chorotype species richness [19]; that is, in each hexagon, we quantified the number of species whose distributions formed part of the chorotype.

6. We then ran a forward-backward stepwise logistic regression using presence/absence of dengue case records as the dependent variable and chorotype variables as predictors. Only the chorotype-variables selected were considered henceforth.

We did not consider primates to be a limiting factor, since dengue cases among humans could be influenced by, but not depend on the presence of primates in the area [28]. Guided by this rationale, we produced the disease favorability model for the late 20th-century cases in two blocks: (1) a stepwise selection of environmental and spatial variables; (2) a later stepwise selection of chorotypes that contribute to improve significantly the model likelihood. In turn, the updating of the model based on cases from the early 21st century consisted of three blocks: (1) forcing the entry of the late 20th-century-model logit equation as a predictor variable; (2) making a later stepwise selection of spatial/environmental variables; (3) ending with a stepwise selection of chorotypes contributing to improve model likelihood.

### Dengue transmission-risk model

We defined a dengue transmission-risk model according to the intersection between a disease model and a vector model. The fuzzy intersection is used to combine models that represent favorable conditions according to limiting factors (i.e., factors that describe imperative conditions for the modelled item to be present) [74]. Thus, the transmission-risk model reflected that, for the pathogen to be transmitted in a given hexagon, two elements must coexist: (1) suitable environmental conditions for disease cases to occur; and (2) suitable conditions for the presence of vectors. In operative terms, the intersection consisted of selecting, for every hexagon, the lowest favorability value among those obtained in the different models [66]. This approach has been used before to reflect the simultaneous need of suitable environments and mammal assemblages for the zoonotic transmission of the Ebola virus to humans [18]. Transmission-risk models were made for both the late 20th and the early 21st centuries. Favorability (F) values were finally reinterpreted as transmission-risk values following this scale: high favorability (i.e., $F > 0.8$ [75]) was referred as high transmission risk; intermediate-high favorability values ($0.5 \leq F \leq 0.8$) were referred as intermediate-high risk; intermediate-low favorability values ($0.2 \leq F < 0.5$) were referred as intermediate-low risk; and low favorability (i.e. $F < 0.2$ [75]) was referred as low transmission risk. $F = 0.2$ and $F = 0.8$ match approximately the

inflection points in the logistic favorability function, while F = 0.5 is the threshold above with the transmission probability defined by spatial and environmental factors is higher than the random transmission probability [51].

## Transmission-model refinement

The early 21st-century dengue transmission-risk model was refined through model enhancing and downscaling. The models for the 21st century described above, based on variables that are also available for the late 20th century, were useful for risk-map comparisons between periods; however, an enhanced early 21st-century model permitted a more updated representation of factors that could aid in defining the risk of disease transmission. Enhancing was performed through the development of new disease and vector models on the basis of an expanded set of predictors, i.e. complementing the former set of variables with others only available for the early 21st century: human population density, infrastructures, land use, vegetation cover, and forest loss (S3 Table). Descriptors of livestock density were also considered for the enhancement of vector models. The proximity of human populations and activities not only imply the availability of human potential hosts. Human-modified environments usually provide chances for the local reproduction of urban *Aedes* mosquitoes (e.g. water points) [76].

The enhanced transmission-risk model was finally downscaled from the initial 7,774-km² spatial resolution to a new grid based on 58,612 hexagons of 2,591 km² (i.e. to 66.7% smaller units), using the direct downscaling approach [77]. Model predictions should remain meaningful after this downscaling has taken place as, according to Bombi & D'Amen [77], predictions are not severely affected by a 10-fold shortening of side lengths in the case of squared spatial units, which is equivalent to a 99% decrease of the surface area. The direct downscaling consisted of projecting the favorability equation that defined the original model to a set of variables considered in the finer-resolution grid of hexagons. In order to avoid local artifacts that could result from this downscaling, we excluded from the downscalled outputs all favorable areas that were not highlighted by the pre-downscaling models.

## Model assessment and validation

Model goodness-of-fit was evaluated according to Chi-square tests. Discrimination capacity was assessed according to the area under the "receiver operating characteristic (ROC)" curve (AUC) [78]. We also assessed classification capacity based on two favorability thresholds: 0.5, at which probability is equal to the overall prevalence [51]; and 0.2, below which the risk of disease transmission was considered to be low (see above). The classification indices employed were the sensitivity, the specificity, the correct classification rate (CCR), Cohen's kappa, the under-prediction rate, and the over-prediction rate [50, 79].

We validated the predictive capacity of the late 20th-century disease and transmission-risk models through the evaluation of their discrimination and classification capacities with regard to the 2001–2017 case record. Similarly, the predictive capacity of the early 21st-century model was validated with regard to the dengue cases reported in 2018 and 2019.

## Contribution of the zoonotic factor

We used a variation partitioning approach [80] to calculate the relative contribution of non-human primates in determining the environmental favorability for the occurrence of dengue cases. We estimated how much of the variation in favorability for the occurrence of dengue cases was explained by the pure effect of primates (here represented by primate chorotypes), and how much was explained by the pure effect of environmental and spatial constraints. The method used [81] also allowed us to calculate how much of the variation in favorability was

attributable to both factors (i.e., the shared effect), because primate ranges, in the same way as disease cases, are likely to be influenced by environmental and spatial constraints.

To map the areas in which the sylvatic cycle could have contributed to increase the record of dengue cases in humans, we identified the hexagons in which: 1) favorability values for the presence of dengue cases were ≥0.2; and 2) the difference between favorability values provided by the dengue model, and favorability values provided by a model not considering chorotypes, was positive and ≥0.1.

## Results

### Vector models

**Urban mosquitoes.** The global distribution of occurrence records and of favorable areas (as defined by environmental and spatial variables) (F≥0.2) for the presence of *Ae. aegypti* and *Ae. albopictus* in the late 20th and the early 21st centuries can be seen in S1 Fig. *Ae. albopictus* was less widespread but showed a more expansive spatial trend. In the late 20th century, favorable areas for *Ae. aegypti* covered extensive regions in North and South America, but included little of the inner Amazon basin. In contrast, *Ae. albopictus* exhibited highly restricted favorable areas in western USA and in South-Brazil coastal areas. In Africa, *Ae. aegypti* occupied large tropical regions, whereas *Ae. albopictus* only occurred in some areas to the south and the north-west of the continent and in Madagascar. Favorable areas for both species were similar in Asia, although they extended further westward for *Ae. aegypti* and eastward for *Ae. albopictus*. There were more favorable areas for *Ae. aegypti* in Australia than for *Ae. albopictus*, but the opposite was the case in New Zealand. In Europe, only *Ae. albopictus* occurred, with favorable areas extending across the Mediterranean region. These models are strongly characterized by the spatial factor, and highlight the environmental relevance of shorter distances to population centers and high annual precipitation (S4 and S5 Tables). The presence of *Ae. aegypti* was also favored by high summer temperatures though *Ae. albopictus* was favored in the temperate-conifer-forest ecoregion by low elevations and a high temperature annual range.

During the early 21st century (S1 Fig), favorable areas for *Ae. aegypti* in America reached most of the Amazon basin and expanded south to Argentina and Chile, as well as into North-West USA. *Ae. albopictus* occupied new areas in North and South America, and spread radially in Central Africa, northward into East Asia, and east and westward in the Mediterranean region of Europe. The models show that both species expanded their spatial/environmentally favorable areas into tropical broadleaf forests and temperate grasslands/savannas (S4 and S5 Tables). The range of *Ae. aegypti* also expanded in temperate conifer forests, and was favored by high winter temperatures. *Ae. albopictus* spread in the Mediterranean and in the temperate-broadleaf ecoregions, its presence favored by a high precipitation seasonality. For both species, the refined models outlined the relevance of human presence in explaining the ongoing spread: high human population density, intensive livestock rearing, and, for *Ae. albopictus*, the proximity of railways and roads.

**Sylvatic mosquitoes.** Details of the favorability models generated for the five sylvatic mosquito species are shown in S2 Fig and S6 and S7 Tables. The presence of the four continental species, namely *Ae. africanus*, *Ae. luteocephalus*, *Ae. niveus*, and *Ae. vittatus*, is favored by high minimum temperatures in the coldest months, and in some cases also by high maximum annual temperatures or high precipitation seasonality. *Aedes polynesiensis* was only characterized by its Pacific-insular spatial pattern. Some tropical ecoregions, linked to moist broadleaf forests or to grasslands and savannas, favor the presence of the African and Asian endemics. In contrast, the old-world species *Ae. vittatus* finds suitable habitats in Mediterranean landscapes as well, especially close to human-populated regions. The refined models highlight the

relevance of humanized environments including the presence of croplands, areas inhabited by livestock or humans, and human infrastructures.

**Integrated vector models.** Favorable areas for the presence of at least one dengue-vector species, as outlined by the fuzzy union of all the single-species outputs, suggest key differences across the two centuries in South America, where spatial/environmentally favorable areas have spread, and in Australia, where favorability values have decreased (Fig 2). In the Mediterranean basin, favorable areas have extended to the Maghreb, and are beginning to spread to the European side.

## Disease models

The distribution of favorable areas (F $\geq$ 0.2) for the presence of dengue cases shows that changes have taken place in the continents since the late 20th century (Fig 2). Favorable areas for dengue have spread southward in South America, inland in the Amazon, eastward in Africa, and to the south-west in Asia. Favorability values have also increased in South-East Asia, North Australia, and Papua New Guinea. The refined model also outlined favorable areas in Japan and South Korea (Figs 3 and S3). Europe, a dengue-free continent in the late 20th century, is currently showing favorable areas in the south, among which are a rising number of urban locations.

The proximity to population centers was the most significant predictor in the model that described areas favorable to the occurrence of dengue cases during the late 20th century (S8 Table). Dengue was favored in a variety of tropical ecoregions including forests and savannas, mangroves, montane grasslands, and xeric lands, as well as by low elevations and a high minimum temperature in the coldest month (S8 Table). Increasing favorability during the early 21st century occurred outside the tropical regions in temperate grasslands, and in areas with high maximum annual temperatures and high pluviometric irregularity but low annual temperature ranges and rainfall. As for the vectors, the refined disease model reaffirms the relevance of human presence. Primate chorotypes contributed significantly to all these models (see below).

## Dengue transmission-risk models

Differences between the late 20th-century disease and transmission-risk models are most visible in South America (Fig 2A), where a vast area along the north-western coasts, and around rivers crossing the Amazon basin, were favorable for the presence of dengue and unfavorable for the presence of vectors. This was also the case in the Horn of Africa and in the north of Papua New Guinea. In contrast, in the early 21st century, this pattern was only seen in Peru, Bolivia, and Argentina, as well as in Saudi Arabia and Iraq (Fig 2B). The refined transmission-risk model for the early 21st century (Fig 3) still indicated significant risk areas in Japan, South Korea, and some European cities.

## Model assessment

**Model evaluation.** The AUC values of all vector, disease, and transmission-risk models were >0.925 (Table 1), pointing to "outstanding" discrimination capacities according to Hosmer and Lemeshow [82], although this could be a result of the worldwide geographic extent of the calibration area. For a favorability threshold of 0.5, the CCR ranged between 0.771 and 0.884. Kappa values ranged between 0.228 and 0.252 in all the 20th-century models, and ranged between 0.352 and 0.544 in the early 21st-century models. Nevertheless, in the disease models and transmission-risk models for the 21st century, Kappa was always $\geq$0.518.

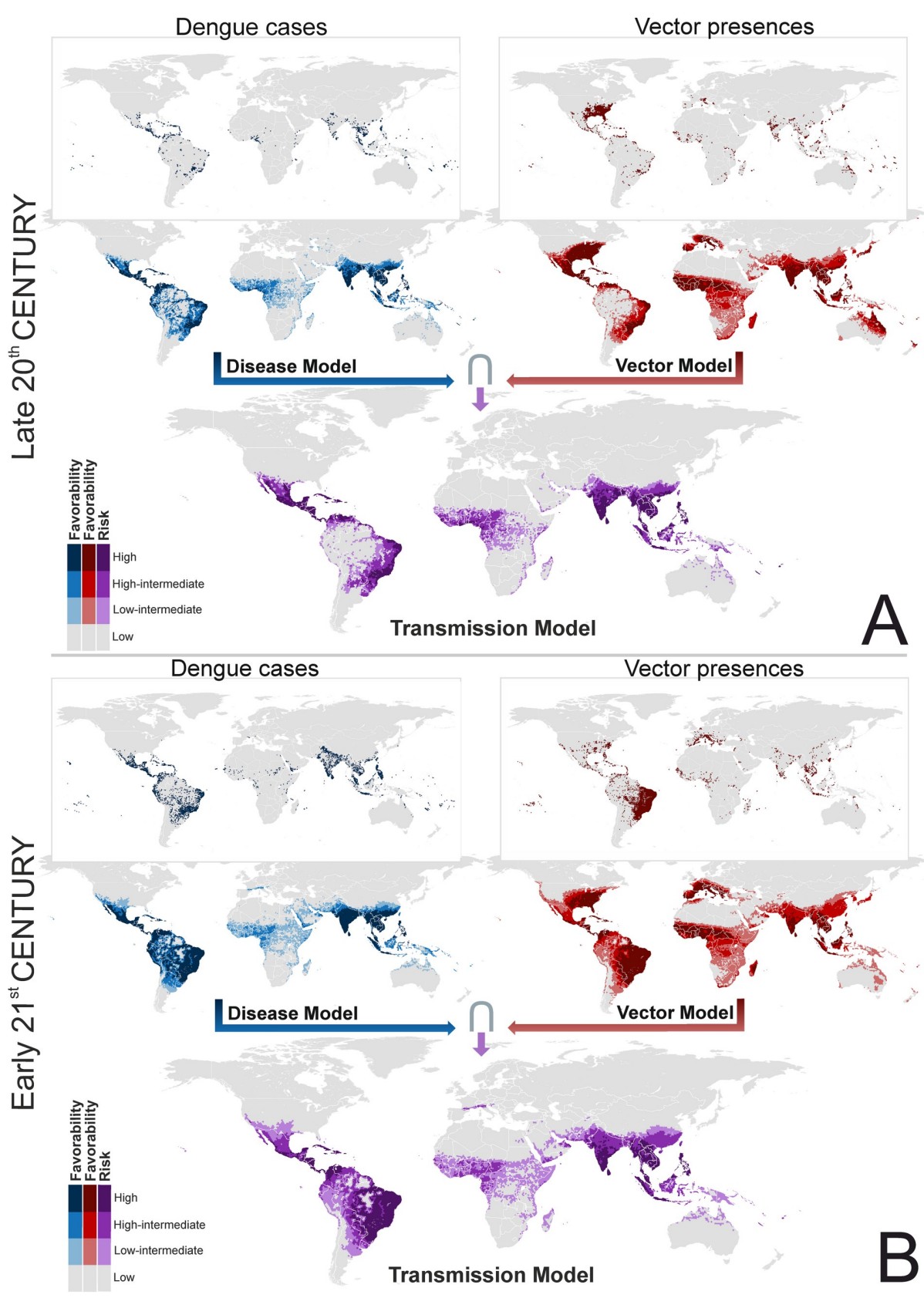

**Fig 2. Global disease, vector and transmission-risk models.** A: maps for the late 20[th] century. B: maps for the early 21[st] century. The risk of transmission is estimated as the intersection (∩) between favorable conditions for the occurrence of dengue cases and favorable conditions for the presence of vector species. The spatial resolution is based on 7,774-km² hexagons. Recorded occurrences of dengue cases and vector presences are also mapped. Coast lines source: https://developers.google.com/earth-engine/datasets/catalog/FAO_GAUL_2015_level0.

The likelihood of underestimating the degree of favorability in areas where vectors and dengue cases occurred was low, as denoted by sensitivity values >0.800 (i.e., >80% of recorded presences were classified in favorable areas), and by under-prediction values <0.025 (i.e., less than 2.5% of the unfavorable spatial units showed recorded presences).

Compared to the 0.5-favorability threshold, when a 0.2 threshold was adopted, the CCR values of all models decreased by an average 12.67% (SD = 3.85), and kappa values decreased by

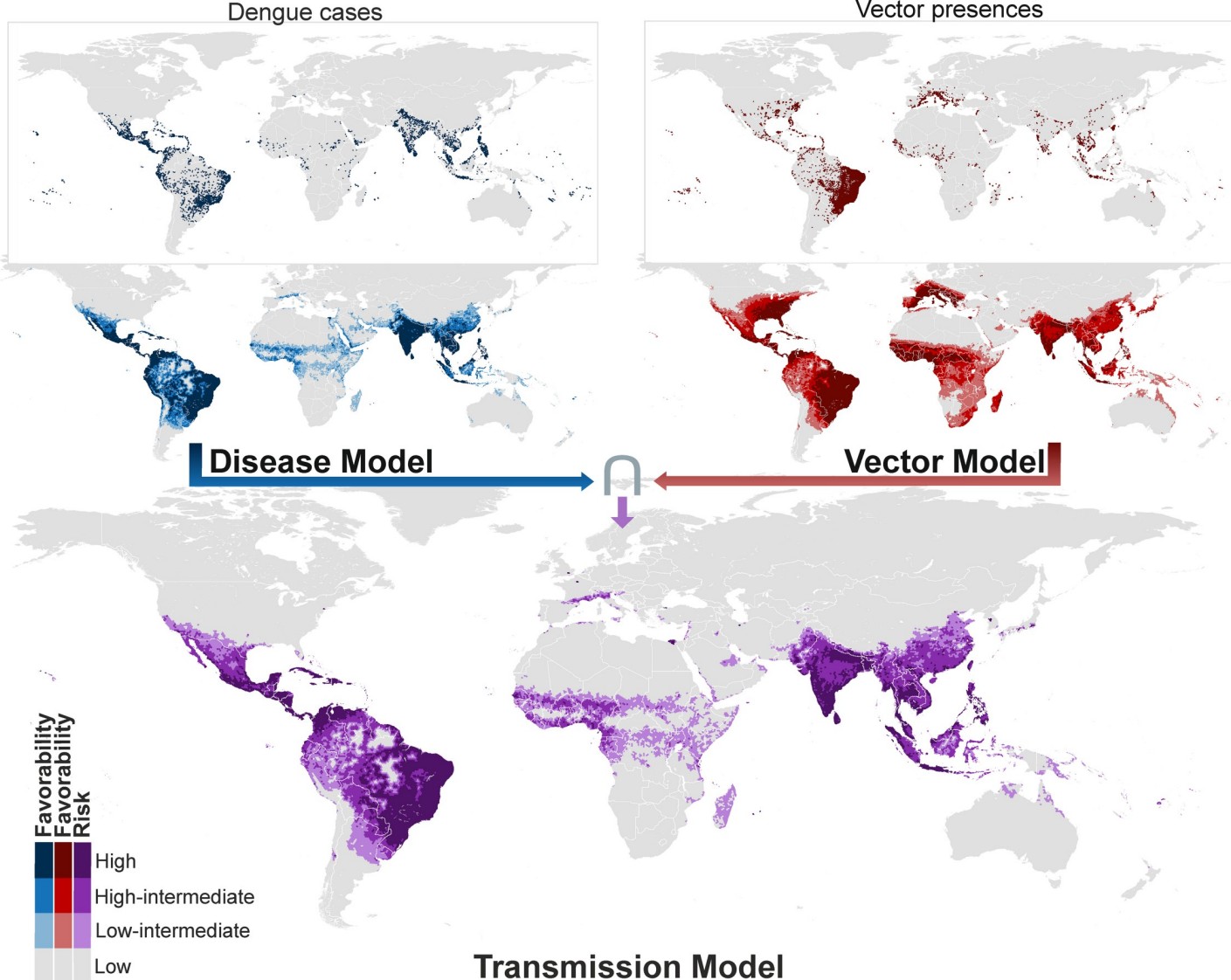

**Fig 3. Refined global disease, vector and transmission-risk models for the early 21[st] century.** The risk of transmission is estimated as the intersection (∩) between favorable conditions for the occurrence of dengue cases and favorable conditions for the presence of vector species. Compared to the models in Fig 2, additional predictor variables only available for the 21[st] century were considered, and the spatial resolution was based on 2,591-km² hexagons. Recorded occurrences of dengue cases and of vector presences are also mapped. See pre-downscaled versions of these models in S4 Fig. Coast lines source: https://developers.google.com/earth-engine/datasets/catalog/FAO_GAUL_2015_level0.

**Table 1. Model assessment based on discrimination and classification capacities respect to vector and disease records of the same period.** AUC: area under the receiver operator characteristic curve; FCT: favorability classification threshold; Kappa: Cohen's kappa; Sens.: sensitivity; Spec.: specificity; CCR: correct classification rate; Underp.: underprediction rate; Overp.: overprediction rate.

| | MODEL | AUC | FCT | Kappa | Sens. | Spec. | CCR | Underp. | Overp. |
|---|---|---|---|---|---|---|---|---|---|
| Late 20th century | Vector | 0.935 | 0.5 | 0.228 | 0.936 | 0.762 | 0.771 | 0.005 | 0.825 |
| | | | 0.2 | 0.150 | 0.990 | 0.637 | 0.655 | 0.001 | 0.872 |
| | Disease | 0.934 | 0.5 | 0.234 | 0.893 | 0.851 | 0.852 | 0.004 | 0.838 |
| | | | 0.2 | 0.164 | 0.964 | 0.767 | 0.773 | 0.001 | 0.882 |
| | Transmission risk | 0.927 | 0.5 | 0.252 | 0.822 | 0.876 | 0.874 | 0.007 | 0.823 |
| | | | 0.2 | 0.178 | 0.926 | 0.794 | 0.798 | 0.003 | 0.873 |
| Early 21st century | Vector | 0.926 | 0.5 | 0.352 | 0.925 | 0.760 | 0.777 | 0.011 | 0.704 |
| | | | 0.2 | 0.210 | 0.991 | 0.580 | 0.620 | 0.002 | 0.795 |
| | Disease | 0.948 | 0.5 | 0.518 | 0.903 | 0.859 | 0.864 | 0.013 | 0.564 |
| | | | 0.2 | 0.359 | 0.980 | 0.730 | 0.757 | 0.003 | 0.696 |
| | Transmission risk | 0.939 | 0.5 | 0.533 | 0.822 | 0.888 | 0.881 | 0.024 | 0.531 |
| | | | 0.2 | 0.375 | 0.965 | 0.749 | 0.772 | 0.006 | 0.684 |
| Early 21st century (refined) | Vector | 0.935 | 0.5 | 0.369 | 0.942 | 0.768 | 0.785 | 0.008 | 0.693 |
| | | | 0.2 | 0.241 | 0.991 | 0.622 | 0.658 | 0.002 | 0.778 |
| | Disease | 0.956 | 0.5 | 0.531 | 0.903 | 0.866 | 0.870 | 0.013 | 0.552 |
| | | | 0.2 | 0.386 | 0.981 | 0.752 | 0.777 | 0.003 | 0.678 |
| | Transmission risk | 0.944 | 0.5 | 0.544 | 0.830 | 0.891 | 0.884 | 0.022 | 0.522 |
| | | | 0.2 | 0.418 | 0.955 | 0.784 | 0.803 | 0.007 | 0.652 |

an average 31.04% (SD = 4.90), which is related to the average 14.91% (SD = 4.25) decrease observed in the specificity values (i.e., some areas without presence records were shown by the models to increase in favorability). Nevertheless, the 0.2-threshold also minimized the likelihood of underestimating the degree of favorability in areas where vectors and dengue cases have occurred, as it produced an approximately 10% increase of the sensitivity values, and an approximately 75% decrease of the under-prediction values.

**Predictive capacity.** The late 20th-century disease and transmission-risk models demonstrated meaningful predictive capacities with respect to the early 21st-century dengue-case records. In many aspects, the assessments provided better results when the observations compared to the models were "future" cases than when we used for comparison the sets of records employed for model training (see Tables 1 and 2, and S9). The AUC values were always >0.910. Considering both the 0.5 and the 0.2-favorability thresholds, and compared to the above model evaluation, the kappa values of the disease and transmission-risk models increased by an average of 59.4% (SD = 5.1) when assessed with respect to all dengue cases reported during 1970–2017, which is similar to the 57.6% increase (SD = 5.1) when assessed with respect to the 2001–2017 cases alone. The CCR also experienced an average 4.6% increase (SD = 1.8) with respect to its evaluation values. This improvement was related to an average 5.8% increase (SD = 0.8) in model specificity, which was always >0.900 with the 0.5 favorability threshold (Table 2) and >0.820 with the 0.2 threshold. Sensitivity values experienced an average 14.3% decrease (SD = 4.9). Nevertheless, sensitivity was always >0.670 with the 0.5 threshold and >0.820 with the 0.2 threshold. Finally, the underprediction rate decreased by an average of 56.5% (SD = 13.8), which indicates that many favorable areas free from disease during the late 20th century experienced outbreaks after 2000.

The early 21st-century (2001–2017) models also showed meaningful predictive capacities (Table 2). Compared to the above model assessment (referenced to the 2001–2017 data), when the whole 2001–2019 period was considered, the kappa values increased by an average of 5.6%

**Table 2. Validation of model predictive capacity based on discrimination and classification performance respect to disease records of a later period.** AUC: area under the receiver operator characteristic curve; FCT: favorability classification threshold; Kappa: Cohen's kappa; Sens.: sensitivity; Spec.: specificity; CCR: correct classification rate; Underp.: underprediction rate; Overp.: overprediction rate.

| | MODEL | Records of reference for validation purposes | AUC | FTC | Kappa | Sens. | Spec. | CCR | Underp. | Overp. |
|---|---|---|---|---|---|---|---|---|---|---|
| **Late 20th century** | Disease | 1970 to 2017 | 0.925 | 0.5 | 0.558 | 0.778 | 0.905 | 0.891 | 0.031 | 0.486 |
| | | | | 0.2 | 0.463 | 0.888 | 0.826 | 0.833 | 0.017 | 0.604 |
| | Transmission risk | | 0.915 | 0.5 | 0.535 | 0.677 | 0.923 | 0.895 | 0.043 | 0.470 |
| | | | | 0.2 | 0.465 | 0.821 | 0.848 | 0.845 | 0.026 | 0.590 |
| | Disease | 2001 to 2017 | 0.923 | 0.5 | 0.538 | 0.781 | 0.901 | 0.888 | 0.028 | 0.514 |
| | | | | 0.2 | 0.440 | 0.888 | 0.820 | 0.827 | 0.016 | 0.627 |
| | Transmission risk | | 0.914 | 0.5 | 0.515 | 0.678 | 0.918 | 0.892 | 0.040 | 0.501 |
| | | | | 0.2 | 0.445 | 0.823 | 0.843 | 0.841 | 0.025 | 0.614 |
| **Early 21st century** | Disease | 2001 to 2019 | 0.948 | 0.5 | 0.548 | 0.898 | 0.867 | 0.870 | 0.015 | 0.530 |
| | | | | 0.2 | 0.385 | 0.979 | 0.737 | 0.765 | 0.004 | 0.671 |
| | Transmission risk | | 0.939 | 0.5 | 0.557 | 0.813 | 0.894 | 0.884 | 0.027 | 0.498 |
| | | | | 0.2 | 0.402 | 0.962 | 0.756 | 0.780 | 0.007 | 0.658 |
| | Disease | 2018 and 2019 | 0.934 | 0.5 | 0.296 | 0.931 | 0.817 | 0.823 | 0.005 | 0.780 |
| | | | | 0.2 | 0.186 | 0.987 | 0.689 | 0.705 | 0.001 | 0.850 |
| | Transmission risk | | 0.915 | 0.5 | 0.305 | 0.831 | 0.847 | 0.846 | 0.011 | 0.768 |
| | | | | 0.2 | 0.198 | 0.978 | 0.708 | 0.722 | 0.002 | 0.843 |
| **Early 21st century (refined)** | Disease | 2001 to 2019 | 0.957 | 0.5 | 0.561 | 0.898 | 0.873 | 0.876 | 0.015 | 0.517 |
| | | | | 0.2 | 0.414 | 0.979 | 0.759 | 0.785 | 0.004 | 0.651 |
| | Transmission risk | | 0.944 | 0.5 | 0.564 | 0.816 | 0.896 | 0.887 | 0.026 | 0.491 |
| | | | | 0.2 | 0.445 | 0.95 | 0.791 | 0.810 | 0.008 | 0.625 |
| | Disease | 2018 and 2019 | 0.945 | 0.5 | 0.31 | 0.943 | 0.824 | 0.830 | 0.004 | 0.771 |
| | | | | 0.2 | 0.202 | 0.989 | 0.710 | 0.725 | 0.001 | 0.841 |
| | Transmission risk | | 0.924 | 0.5 | 0.31 | 0.834 | 0.849 | 0.848 | 0.011 | 0.765 |
| | | | | 0.2 | 0.224 | 0.969 | 0.742 | 0.754 | 0.002 | 0.827 |

(SD = 1.2), and the CCR values increased by 0.8% (SD = 0.3) (Tables 1, 2 and S9). When only the 2018 and 2019 data were employed, both kappa and CCR values decreased, but they were always >0.290 and >0.820, respectively, with the 0.5-favorability threshold, and >0.180 and >0.700, respectively, with the 0.2 threshold (Table 2).

## Contribution of the sylvatic cycle

A total of 51 chorotypes were detected: 24 chorotypes in Asia, 13 in Africa, and 14 in America (S5–S7 Figs). Thus the early 21st-century disease model is an update of the late 20th-century disease model (see Fig 1), as all chorotypes in the latter are also included in the former. Taking this into account, Asia contributed to the late 20th-century model with two chorotypes, including species in the following genera: *Hylobates*, *Trachyphitecus*, *Nomascus*, and *Pygathrix* in chorotype AS8; and *Hylobates*, *Presbytis*, *Nycticebus*, and *Trachypithecus* in chorotype AS15. Four additional Asian chorotypes were included in the 21st-century model, with the following species: *Macaca* in chorotype AS5; *Hylobates*, *Presbytis*, *Symphalangus*, and *Nycticebus* in chorotype AS7; *Loris*, *Semnopithecus*, and *Macaca* in chorotype AS9; and *Carlito* in chorotype AS19. The African chorotype AF2, with the genera *Arctocebus*, *Cercopithecus*, *Colobus*, *Euoticus*, *Gorilla*, *Lophocebus*, *Mandrillus*, *Miophitecus*, and *Sciurocheirus*, was included in the 20th-century model, whereas no additional African chorotype was included in the 21st-century model. South America contributed to the 20th-century model with three chorotypes, including species in the following genera: *Alouatta*, *Sapajus*, *Brachyteles*, *Callithrix*, *Callicebus*, and

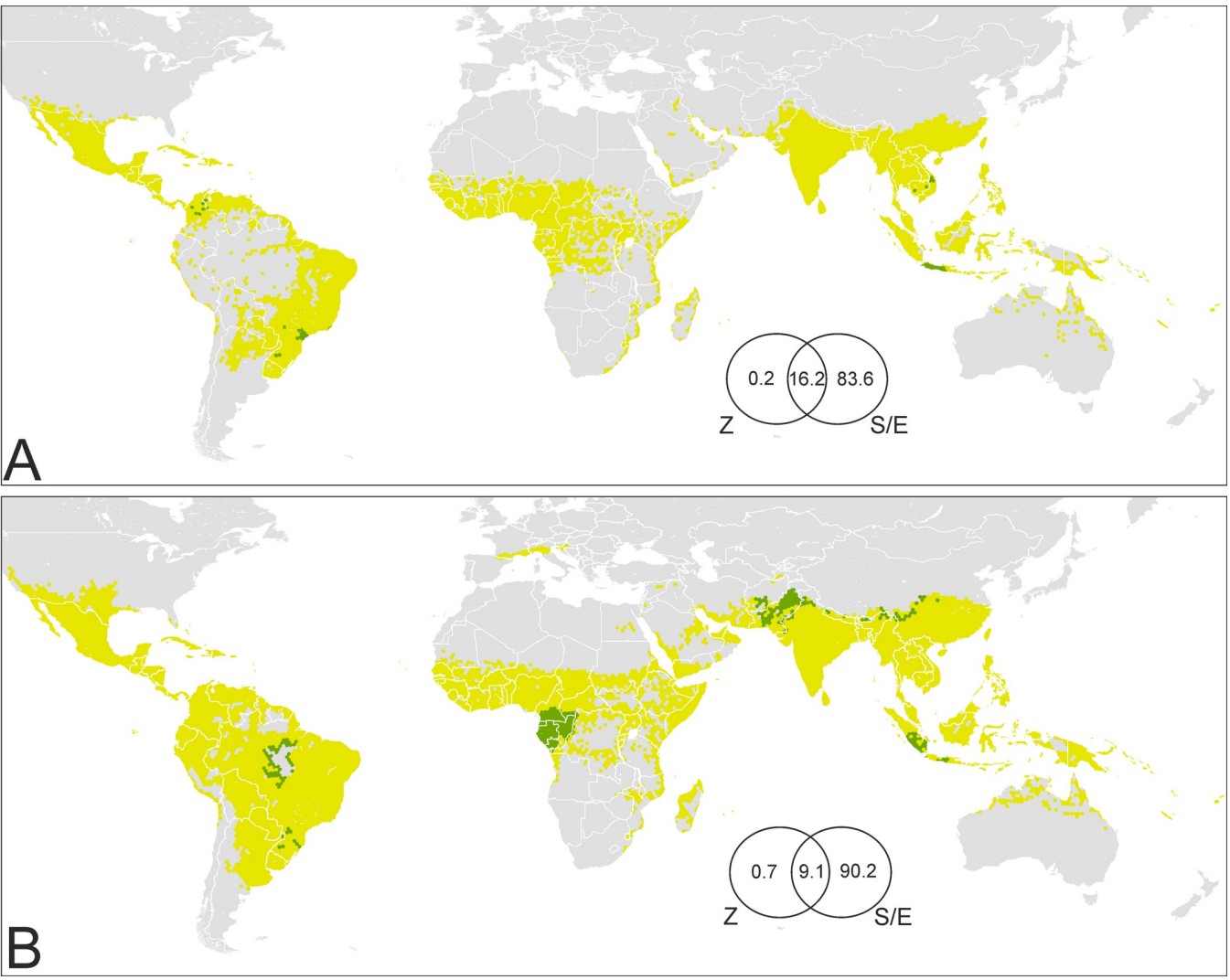

**Fig 4. Areas of potential influence of sylvatic cycles on the presence of dengue in humans.** (A) Late 20th century; (B) early 21st century. Green: >0.1 increase of favorability values attributed to primate chorotypes; yellow: ≤0.1 increase of favorability values attributed to primate chorotypes; grey: area with low risk of dengue transmission. Venn diagrams: The numbers are percentages of contribution to the distribution of favorability in the disease models (Z: Zoogeographic factor; S/E: Spatial/Environmental factor). Coast lines source: https://developers.google.com/earth-engine/datasets/catalog/FAO_GAUL_2015_level0.

*Leontophitecus* in chorotype SA4; *Aotus, Cebus, Ateles,* and *Saguinus* in chorotype SA5; and *Aotus, Saguinus, Oreonax,* and *Callicebus* in chorotype SA14. An additional South-American chorotype, SA2, was included in the 21st-century model, with species of the genera *Alouatta, Ateles, Callicebus, Chiropotes,* and *Mico* (all the variables included in the disease models can be seen in S8 Table).

Primate chorotypes contributed to explain a maximum of 16.4% of the variation in favorability for the presence of dengue in the late 20th century (Fig 4A). However, only 0.2% of the variation can be exclusively attributed to these chorotypes. The remaining 16.2% of the variation was indistinguishably attributed to chorotypes and to spatial/environmental factors, as the distribution of primate ranges is also dependent on the environment. In the early 21st-century model, chorotypes contributed a maximum of 9.8% to explain the variation in favorability, although only 0.7% could be exclusively attributed to them (Fig 4B).

The distribution of areas in which there was a >0.1 favorability increase, as an exclusive effect of primates, is shown in Fig 4. In the late 20th century, these areas were located in Java (Indonesia), in some areas of Cambodia and Vietnam, in northern Colombia, and in southern Brazil (Fig 4A). In the early 21st century, the possible contribution of primate chorotypes expanded to Sumatra in Indonesia, and also involved Asian areas of Afghanistan, Pakistan, India, Nepal, and China in Asia, Amazonian areas of Brazil, and some African countries in the western Congo basin, mainly Cameroon, Gabon, Equatorial Guinea, and the Republic of Congo (Fig 4B).

## Discussion

Our pathogeographic approach is the first to explicitly generate a high-resolution analysis of the geographic changes experienced in the dengue-transmission risk since the late 20th century. During the past century, dengue cases have been reported across a wide range of tropical ecoregions. Based on our research findings, we suggest that areas at risk of dengue transmission included regions in which cases only started being reported after 2000. We show that the distributions of *Aedes aegypti* and *Ae. albopictus* were principally linked to human presence in lowland tropical areas, although *Ae. albopictus* started to occur in some temperate regions as well. In the current century, dengue-risk areas continue to spread, reflecting the fact that both *Aedes* species are expanding their ranges into a number of temperate ecoregions worldwide. Our study is useful as a basis for suggesting specific management strategies according to the spatial distribution of factors favoring risk, and is the first to take into account the potential contribution of primate biogeography and sylvatic vectors in increasing the risk of dengue transmission.

In certain areas in South Asia that were free from dengue two decades ago, such as Pakistan, the presence of dengue and the occurrence of vectors were environmentally favored; thus, we predict the risk of dengue transmission in those areas. The early 21st-century disease reports strongly confirm this forecast (Fig 5A). In contrast, in the Amazon basin, a successful forecast for the near future was provided by the disease model, but the same was not true of the transmission-risk model, which suggested that dengue presence, but not vector presence, was favored by the environment (Fig 5B). Hence, the Pakistani and Amazonian scenarios would require different prevention strategies. The risk in Pakistan was evident, so ensuring a close microbiological and epidemiological surveillance would have been reasonable (e.g., in the presence of clinically compatible cases, dengue should be suspected and microbiologically confirmed). In the Amazon, meanwhile, the arrival of invasive vectors should have been prevented, but now *Ae. aegypti* and *Ae. albopictus* occur near rivers and tributaries across the basin (Fig 5B). Predicting the establishment of invasive species in new areas is difficult. The dispersal of *Aedes* is strongly influenced by travel and trade routes [37, 39], as much as by the worldwide propagation of pathogens [39, 83]. The progressive spread of invasive *Aedes* species into temperate ecoregions could also be influenced by climate change [9, 84, 85]. However, this situation is further aggravated if anthropogenic factors affect their evolutionary and consequently adaptive potential [41].

The predictive power of our late 20th-century models can be assumed for the early 21st-century models as well, as all of them were derived from the same method. The predictive capacity of the 21st-century models has been confirmed by the reported occurrence of autochthonous dengue after 2017 in Muscat (Oman) [86], Kyoto, and Nara (Japan) [87], and in Spanish coastal cities [88, 89] (S3 Fig). The early 21st-century transmission-risk models predict a spread of the risk in still barely affected areas exposed to the presence of invasive *Aedes*. This is particularly relevant in South-East China, but also in Papua New Guinea, North Australia, South

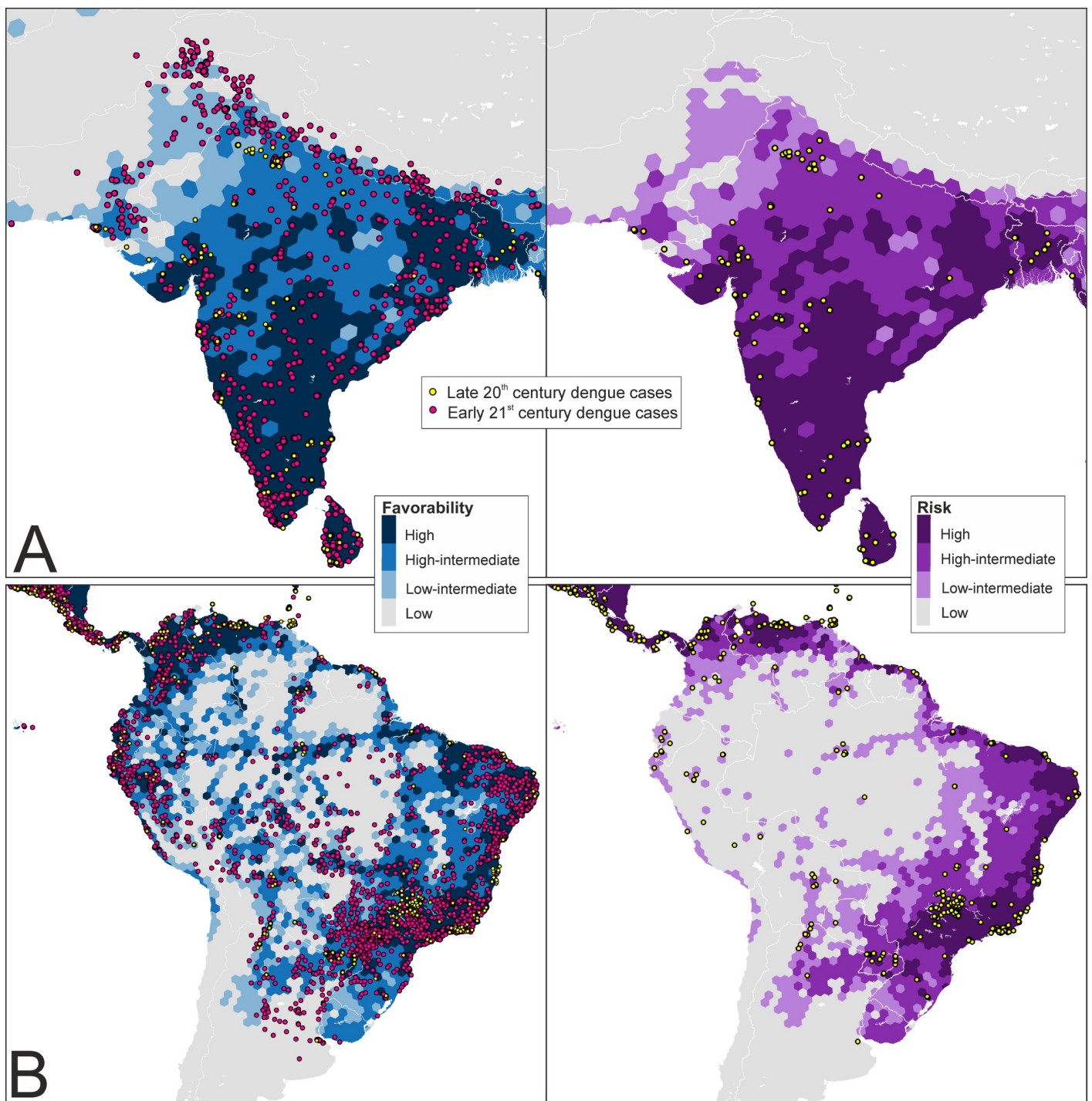

**Fig 5. Late 20<sup>th</sup> century disease and transmission-risk models in the Indian peninsula (A) and South America (B).** These models were calibrated according to human-dengue cases from the late 20th century (Fig 2A). The locations of dengue cases recorded in the late 20th and the early 21st centuries are shown in order to illustrate the predictive capacity of these models (see explanations and implications in the main text). See early 21st-century models and data for these areas in S8 Fig. Coast lines source: https://developers.google.com/earth-engine/datasets/catalog/FAO_GAUL_2015_level0.

USA, the interior regions of Colombia and Venezuela, Madagascar, and, according to the refined model, also in Japan and urban areas of South and Central Europe. Our results suggest that dengue could spread into areas of Argentina and South-West Asia (from Pakistan to the

Arabian Peninsula) where invasive *Aedes* species occur but are scarcely reported (see S8 Fig). In addition, populated areas in Chile, Iran, Iraq, and the Maghreb, still free from invasive *Aedes*, exhibit favorable conditions for the occurrence of vectors and disease. The imminent risk in these locations, despite their distance from the dengue-native regions, should not be discounted given the high level of global connectivity [39] and the influence of human-population density on the intensity of dengue transmission [13]. Relevant precedents are the current expansive trend of autochthonous dengue in the Mediterranean cities [89], and a local transmission reported in New York (USA) [90] that was predicted two years previously [91]. The northern coasts of Chile now have a similar situation to what has been seen in the Amazon basin over the past century; vectors have not yet arrived there but the area—which is environmentally similar to the dengue-affected coasts of Peru—is favorable to the presence of dengue cases. Management policies to prevent the arrival of invasive mosquitoes should be strongly encouraged. Finally, the eastern half of the USA and large sections of South-Saharan Africa exhibit unfavorable conditions (in spatial and/or environmental terms) for dengue even though these areas exhibit some limited favorable conditions and are home to *Aedes* mosquitoes. Thus, measures to be taken should depend on the socioeconomic and environmental conditions of the region. In eastern USA, international travellers should be educated about the threat of mosquito-borne diseases and on the importance of using repellents in endemic areas in order to prevent this region from becoming a spatially favorable zone for local transmission. In Africa, microbiological and epidemiological surveillance should be encouraged and, when needed, internationally supported.

Any distribution modelling approach is subject to limitations primarily derived (1) from the spatio-temporal dynamism of the modelled facts, (2) from uncertainties in the quality of the available information, and (3) from the interpretation of patterns based on correlations between dependent and independent variables (i.e., correlative approaches). First, a high spatio-temporal dynamism affects the distribution of dengue cases and *Aedes* mosquito species. Because of this, the transmission risk in areas that have been favorable for dengue in the past might not always be highlighted by our models. Chances for the disease to reach areas with similar environmental conditions might be different, conditioned by the geographical proximity of vectors and pathogens, i.e. because of the spatio-temporal autocorrelation. We took this autocorrelation into account by considering the spatial factor in the set of predictor variables. Consequently, these models were designed for specific contexts in the spatio-temporal dimension, and so they should be interpreted as focused on the current historical moment. Second, a low quality in the data set might have been a serious drawback in our models if the distribution of false absences were biased with respect to the gradient of environmental conditions, and also if the modelling method used were susceptible to overfitting. One of the methods we employed to addres this problem was the grid approach, as it reduced to a large extent the proportion of area considered to be free from dengue and vectors in the database. In addition, overfitting does not characterize our methodological approach [92], as was confirmed by the fact that 11–42% of the "absence" hexagons were predicted to be favorable by the models (see specificity in Table 1). In any case, some bias could occur in poorly sampled regions, for example in Africa [93], where model predictions should be interpreted with caution. Finally, a cautiounary approach is always advised when using correlative methods. Measures can be taken to avoid multicollinearity and type I errors, but the link between observed covariances and cause-effect relations always depends on the robustness of the a-priori hypotheses supporting the predictors data set. We were careful in this respect, but we still found a little artifact derived from the model downscaling to smaller hexagons. This procedure led us to estimate a high risk of dengue transmission in some populated cities that are geographically distant from the areas highlighted by the pre-downscaling model. We corrected this artifact, so that the final output

ensured a total correspondence between models before and after the downscaling procedure (see Figs 3 and S4).

Our model for defining areas at risk of dengue transmission is broadly similar to those produced in previous studies. However, some differences are worth highlighting. We focus on Messina et al.'s research in 2019 [9], as it provides an update of previous maps [5, 94] and, as we do here, takes into account the distribution of suitable areas for vectors. There are three main methodological differences that could explain discrepancies between our outputs and previous maps: (1) the treatment given to the temporal dimension, (2) the assumptions made for including vector distributions in the models, and (3) the application of a different modelling method (i.e., the logistic regressions and the Favorability Function).

Our models provide perceptions of current trends such as the spread of dengue in the Amazon basin and southern Asia, resulting from the temporal stratification. In addition, our models were trained with cases reported up to 2017, whereas Messina et al. [9] only considered cases up to 2015 and excluded the autochthonous cases that occurred in Europe. This could explain the differences for Europe, Argentina, and Uruguay. We suggest the presence of a high transmission risk in southern France and northern Italy. In South America, as predicted by our models, recent reports demonstrate a significant risk in central Argentina, Peru, Bolivia, Paraguay, and southern and North-West Uruguay (PAHO in www.paho.org; ECDC in ecdc. europa.eu reports), where Messina et al.'s predictions only suggest a modest increase acros the century as a result of climate-warming projections.

The high risk suggested by Messina et al. [9] for the eastern half of the USA is a major difference with our transmission-risk map, as risks in that area depend only on the presence of *Aedes* mosquitoes (compare vector and disease models in Fig 3). The way vector species are integrated in a risk model reflects the *a priori* assumptions that are adopted with respect to these vectors' role in the pathogen transmission. In the case of dengue, it depends entirely on mosquitoes, and so it seems reasonable to adopt an intersection approach in which the risk points to areas favorable to both pathogen and vectors. We did this, and Messina et al. also exclude all areas environmentally unsuitable for vectors from their map [9]. However, their model also considered vectors as part of the predictor-variable ensemble, and this allowed the environment and the vector presence to counter-balance each other with no limiting-factor concerns [74]. We believe that this is justified, because the mere presence of vectors is a risk factor [37, 95]. However, this fact is sufficiently highlighted by a vector model, and the ability to differentiate between factors favoring risk is conductive to evaluating urgency and designing prevention strategies, as seen above. Our approach highlights transmission risks in areas in which both the vector and the disease are environmentally favored, but we also suggest that, regardless of the existence of vector reports, the presence of favorable environments for infections to occur should sound a warning in the event of unprecedented autochthonous cases. These situations, such as that of the Amazon in the late 20[th] century, are only detected by disease models that are kept "blind" to the vector factor during its training phase.

Lastly, our models show particularities that could result from the performance of the algorithms employed. The most glaring case is related to risk predictions involving the entire Mexican territory, whereas these risks are limited to the coasts in previous models [9]. The trend of dengue cases reported in Mexico after 2000 suggests an inland-spread of favorable areas.

In Asia, Africa, and South America, the areas prone to risk of zoonotic transmission to humans—according to our models—largely overlap with dengue transmission-intensity hotspots [13]. Although human-to-human transmission in urban contexts represents the most important virus cycle from the epidemic point of view [26], zoonotic transmission from other primates has also occurred in tropical regions in Asia and Africa [28], suggesting that the medical relevance of forest cycles is, perhaps, underestimated. This means that active disease

surveillance, such as that employed in Brazil for the yellow fever [96], could be misused. According to our results, sylvatic dengue cycles account for a small percentage of the global extent of the human case record, but could be meaningful in sanitary terms in some tropical areas.

The Asian areas with recorded transmissions of forest-dengue serotypes to humans are located in peninsular Malaysia [97, 98] and Borneo [99, 100]. Besides, positive serological responses to the dengue virus have been detected in non-human primates from Indonesia, the Philippines, Cambodia, Vietnam, Malaysia and Thailand [98, 101–103]. The maps presented here suggest that non-human-primate distributions may increase the environmental favorability for the presence of dengue cases in Indonesia (Java and Sumatra), Cambodia, and Vietnam (Fig 4). The fact that these areas overlap with those with seropositivity in non-human primates, and are only approximately 500 km away from locations of confirmed forest dengue in humans, endorses our outputs. Serological surveys and experiments point to the primate genera *Presbytis* and *Macaca*—which are widely represented in the chorotypes involved in our disease models (S5 Fig)—as dengue reservoirs and amplification hosts, suggesting that other areas in Asia could be undiscovered foci of zoonotic dengue transmission [28]. This could be the case for Pakistan, Afghanistan, northern India, Nepal, and China, all inhabited by the genus *Macaca* and here outlined as areas of zoonotic transmission risk (Fig 4B).

In Africa, human infections by a forest dengue serotype were detected in 1966 in Ibadan, Nigeria [56], approximately 1,000 km from the areas where sylvatic cycles could amplify the risk of dengue transmission according to our models: Cameroon, Equatorial Guinea, Gabon, Congo, and the Democratic Republic of the Congo (Fig 4). The Congo basin, specifically Gabon, could have recently experienced epizootic transmission in non-human primates [104]. The record of humans affected by sylvatic dengue also points to regions in western Africa such as Senegal [105, 106]; additionally, epizooties in primates could have also occurred in Nigeria [107], Senegal [108, 109], and Kenya [110]. Species belonging to the genera *Chlorocebus*, *Erythrocebus*, and *Papio* are considered to be dengue reservoirs or amplification hosts [28]. Species from these genera help to characterize the distribution of dengue cases in Africa, while close relatives from the same tribe (e.g., *Miopithecus* and *Cercopithecus*) inhabit the Congo-basin areas here suggested to be at risk of zoonotic dengue transmission (S6 Fig).

Forest occupancy by human activities is considered to be a driver of disease emergence [111–114], increasing the relevance of sylvatic dengue spillover in tropical regions [115]. In Asia and Africa, the real extent of transmission with an enzootic origin could have been neglected due to the impossibility of discerning between forest and urban serotypes [28]. However, spillback cases with primates acting as reservoirs for urban dengue serotypes could also occur [116], and this might be happening in South America [33, 34]. Seropositivity to the dengue virus has been documented in species from the genus *Alouatta* in north-eastern Argentina [117] and Costa Rica [118], from *Cebus* in Costa Rica [118], and from *Leontopithecus* in southeastern Brazil [119]. Precisely, south-eastern Brazil, specifically the Atlantic forests surrounding Bahia, is highlighted by our model as an area at risk of zoonotic transmission to humans (Fig 4). In this region, the yellow-fever virus shows evolutionary dynamics linked to forest primates [32], and vectors of this virus have shown positivity to the presence of dengue strains [33]. Our model also points to a sylvatic-cycle influence on dengue-case occurrence in the Brazilian Amazon, involving chorotypes that include species of the primate genera *Alouatta* and *Cebus* (Figs 4 and S7).

In conclusion, the human influence on the dispersal of *Aedes* mosquitoes, as much as the adaptive potential of these animals, make environments currently supporting the presence of dengue vectors not represent the range of conditions that might allow them to establish populations. Our vector model predictions should, therefore, be taken seriously when detecting

favorable areas for the presence of invasive *Aedes*, and should be considered to be conservative when neglecting the risk in areas that have already reported pioneer populations. Preventing the arrival of invasive mosquitoes is very important, specially in areas where environmental conditions favor transmission of dengue. If vectors already occur in the area, but virus transmission is not environmentally favored, prevention policies should focus on international-traveller education and microbiological surveillance. Our models are also conservative in mapping the increase of favorability derived from the sylvatic cycle, as we only mapped areas where the contribution of primate chorotypes was not correlated with environmental factors such as presence of tropical forests. Thus, the areas prone to sylvatic dengue transmission could be larger than estimated, mostly in Africa. The concentration of evidence suggest the need for studies that address the occurrence of dengue sylvatic cycles in the Atlantic forest of Brazil and the Amazon. Besides, we suggest that forests in north-western Colombia be investigated for sylvatic cycles, as a chorotype including a *Cebus* species seems to have contributed to the increased risk of dengue transmission in the recent past.

## Supporting information

**S1 Table. Number of dengue case reports and vector occurrences considered in the analyses; and number of presences after point transference to a 7,774-km$^2$ hexagons grid.** See source references in the maintext.
(DOCX)

**S2 Table. Literature used for georeferencing the presence of sylvatic dengue vectors.**
(DOCX)

**S3 Table. Independent predictor variables considered for disease, vector, and transmission-risk modelling.** Some variables were used only in specific models: *20$^{th}$-century models; **refined 21$^{st}$-century models; ***refined 21$^{st}$-century vector models; ****disease models.
(DOCX)

**S4 Table. Vector-model (*Aedes aegypti*) logit equations (i.e., linear combinations of predictor variables that form part of the logistic-regression equations).** Variables in bold letters are mentioned in the results section of the main text. B: variable coefficient; SE: standard error; W: Wald parameter; DF: degrees of freedom; S: statistical significance. Variable codes as in S3 Table.
(DOCX)

**S5 Table. Vector-model (*Aedes albopictus*) logit equations (i.e., linear combinations of predictor variables that form part of the logistic-regression equations).** Variables in bold letters are mentioned in the results section of the main text. B: variable coefficient; SE: standard error; W: Wald parameter; DF: degrees of freedom; S: statistical significance. Variable codes as in S3 Table.
(DOCX)

**S6 Table. Sylvatic-vector-model logit equations (i.e., linear combinations of predictor variables that form part of the logistic-regression equations).** The *Aedes polynesiensis* model was only based on the spatial factor. For the rest of species, an environmental model and a spatial model were intersected. These decisions responded to the geographically restricted character of these species distributions. Variables in bold letters are mentioned in the results section of the main text. B: variable coefficient; SE: standard error; W: Wald parameter; DF: degrees of freedom; S: statistical significance. Variable codes as in S3 Table.
(DOCX)

**S7 Table. Sylvatic-vector refined-model logit equations (i.e., linear combinations of predictor variables that form part of the logistic-regression equations).** The *Aedes polynesiensis* model was only based on the spatial factor. For the rest of species, an environmental model and a spatial model were intersected. These decisions responded to the geographically restricted character of these species distributions. Variables in bold letters are mentioned in the results section of the main text. B: variable coefficient; SE: standard error; W: Wald parameter; DF: degrees of freedom; S: statistical significance. Variable codes as in S3 Table. (DOCX)

**S8 Table. Disease-model logit equations (i.e., linear combinations of predictor variables that form part of the logistic-regression equations).** Variables in bold letters are mentioned in the results section of the main text. B: variable coefficient; SE: standard error; W: Wald parameter; DF: degrees of freedom; S: statistical significance. Variable codes as in S3 Table. (DOCX)

**S9 Table. Percentage of increase in values of the model predictive-capacity assessment (i.e., discrimination and classification performance with respect to disease records of a later period) compared to the descriptive-capacity assessment (i.e., discrimination and classification performance with respect to disease records of the same period).** AUC: area under the receiver operator characteristic curve; FCT: favorability classification threshold; Kappa: Cohen's kappa; Sens.: sensitivity; Spec.: specificity; CCR: correct classification rate; Underp.: underprediction rate; Overp.: overprediction rate. (DOCX)

**S1 Fig. Urban-vector presence records and favorability models.** Coast lines source: https://developers.google.com/earth-engine/datasets/catalog/FAO_GAUL_2015_level0. (DOCX)

**S2 Fig. Sylvatic-vector presence records and favorability models.** Models with temporally stable variables in the short term were used for building models shown in main text Fig 2. Models with variables subject to potential change over time in the short term were used for building models shown in main text Fig 3. Coast lines source: https://developers.google.com/earth-engine/datasets/catalog/FAO_GAUL_2015_level0. (DOCX)

**S3 Fig. Zoomed details of dengue transmission-risk models shown in main text Fig 2 (late 20th and 21st century) and 3 (early 21st century refined).** Coast lines source: https://developers.google.com/earth-engine/datasets/catalog/FAO_GAUL_2015_level0. (DOCX)

**S4 Fig. Pre-downscaling refined global disease, vector, and transmission-risk models for the early 21st century.** The risk of transmission is estimated as the intersection (∩) between favorable conditions for the occurrence of dengue cases and favorable conditions for the presence of vector species. Coast lines source: https://developers.google.com/earth-engine/datasets/catalog/FAO_GAUL_2015_level0. (DOCX)

**S5 Fig. Classification dendrograms of primate distributions in Asia.** Violet rectangles: chorotypes significantly related to the distribution of the late 20th-century dengue cases according to a forward-stepwise logistic regression. Green rectangles: chorotypes significantly related only to the distribution of the 21st-century cases. Chorotypes that were finally included in

disease models are highlighted with a violet asterisk for the late 20[th] century, and with a green asterisk for the 21[st] century. Asian chorotype names are coded as AS1 to AS24. Coast lines source: https://developers.google.com/earth-engine/datasets/catalog/FAO_GAUL_2015_level0.
(DOCX)

**S6 Fig. Classification dendrograms of primate distributions in Africa.** Violet rectangles: chorotypes significantly related to the distribution of the late 20[th]-century dengue cases according to a forward-stepwise logistic regression. Green rectangles: chorotypes significantly related only to the distribution of the 21[st]-century cases. Chorotypes that were finally included in disease models are highlighted with a violet asterisk for the late 20[th] century, and with a green asterisk for the 21[st] century. African chorotype names are coded as AF1 to AF13. Coast lines source: https://developers.google.com/earth-engine/datasets/catalog/FAO_GAUL_2015_level0.
(DOCX)

**S7 Fig. Classification dendrograms of primate distributions in America.** Violet rectangles: chorotypes significantly related to the distribution of the late 20[th]-century dengue cases according to a forward-stepwise logistic regression. Green rectangles: chorotypes significantly related only to the distribution of the 21[st]-century cases. Chorotypes that were finally included in disease models are highlighted with a violet asterisk for the late 20[th] century, and with a green asterisk for the 21[st] century. American chorotype names are coded as SA1 to SA14. Coast lines source: https://developers.google.com/earth-engine/datasets/catalog/FAO_GAUL_2015_level0.
(DOCX)

**S8 Fig. Early 21[st]-century disease and transmission-risk models in the Indian peninsula (A) and South America (B).** These models were calibrated according to human-dengue cases from the late 21[st] century (Fig 3). The locations of dengue cases recorded in the early 21[st] century and from 2018 to 2019 are shown in order to illustrate the predictive capacity of these models. Coast lines source: https://developers.google.com/earth-engine/datasets/catalog/FAO_GAUL_2015_level0.
(DOCX)

## Acknowledgments

We thank Adrían Martín-Taboada for his contribution in the grid-cell design, and Jose M. García-Carrasco for his support with the verification of some predictor variables.

## Author Contributions

**Conceptualization:** Alisa Aliaga-Samanez, Raimundo Real, Marina Segura, Jesús Olivero.

**Data curation:** Alisa Aliaga-Samanez, Marina Cobos-Mayo, Jesús Olivero.

**Formal analysis:** Alisa Aliaga-Samanez, Marina Cobos-Mayo, Jesús Olivero.

**Funding acquisition:** Alisa Aliaga-Samanez, Raimundo Real, Jesús Olivero.

**Investigation:** Alisa Aliaga-Samanez, Marina Cobos-Mayo, Raimundo Real, Jesús Olivero.

**Methodology:** Alisa Aliaga-Samanez, Marina Cobos-Mayo, Raimundo Real, Jesús Olivero.

**Project administration:** Jesús Olivero.

**Resources:** David Romero, Jesús Olivero.

**Software:** Raimundo Real, Jesús Olivero.

**Supervision:** Raimundo Real, Marina Segura, Julia E. Fa, Jesús Olivero.

**Validation:** Alisa Aliaga-Samanez, Jesús Olivero.

**Visualization:** Alisa Aliaga-Samanez, David Romero, Jesús Olivero.

**Writing – original draft:** Alisa Aliaga-Samanez, Marina Cobos-Mayo, Jesús Olivero.

**Writing – review & editing:** Alisa Aliaga-Samanez, Raimundo Real, Marina Segura, David Romero, Julia E. Fa, Jesús Olivero.

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
