## [Decision Letter · Decision Letter 0]

1 Feb 2021

Dear Ms Aliaga,

Thank you very much for submitting your manuscript "Worldwide dynamic biogeography of zoonotic and anthroponotic dengue" for consideration at PLOS Neglected Tropical Diseases. As with all papers reviewed by the journal, your manuscript was reviewed by members of the editorial board and by several independent reviewers. In light of the reviews (below this email), we would like to invite the resubmission of a significantly-revised version that takes into account the reviewers' comments. 

The reviewers generally agreed that there is a need for a more clear explanation of the model methodology and discussion of the limitations of the approaches and underlying datasets.

We cannot make any decision about publication until we have seen the revised manuscript and your response to the reviewers' comments. Your revised manuscript is also likely to be sent to reviewers for further evaluation.

Sincerely,

Michael R Holbrook, PhD

Associate Editor

Elizabeth Carlton

Deputy Editor

Reviewer's Responses to Questions

**Key Review Criteria Required for Acceptance?**

**Methods**

-Are the objectives of the study clearly articulated with a clear testable hypothesis stated?

-Is the study design appropriate to address the stated objectives?

-Is the population clearly described and appropriate for the hypothesis being tested?

-Is the sample size sufficient to ensure adequate power to address the hypothesis being tested?

-Were correct statistical analysis used to support conclusions?

-Are there concerns about ethical or regulatory requirements being met?

Reviewer #1: It is not clear how the model was validated on independent data. The authors draw an arbitrary line at the year 2000 and then define each dataset on either side of this as acceptable to validate results against. Is there a previous method to justify this selection or are the authors biasing their results by making this choice? This choice of validation method needs further justification and it needs to be clearer in their methods and results on how this was done.

Line 121: Remove first sentence. The authors just need to state they used a worldwide grid for the reader to understand the study area is global. 

Lines 123-127: The downscaling methodology reference needs to be included here and preferably only mentioned once in the manuscript. 

Line 130: Not including pre 1970 records is a limitation placed on results. 

Why were models updated by adding variables of risk? What distribution of risk?

Lines 301-303. Why was the lowest favourability value selected for the dengue transmission risk model? This may have resulted in major cities in northern Europe and southern Australia and New Zealand singled out as a risk of transmission, whereas areas throughout these regions were not. Areas surrounding major cities should be just as suitable, as human movement will spread the disease into surrounding areas which, as indicated in Europe in particularly, has a high probable for vector presence. Whereas the Californian San Fernando Valley, a very dry and cold area currently, is predicted to increase in risk over large areas. As such I do not think the transmission model in Figure 3 truly represents the risk of transmission, especially considering the model supposedly takes into consideration all human settlements and authors should justify their choice of selecting the lowest favourability value.

Vector favourability

It should be made clearer by the authors that this model does not accurately estimate the favourability for vectors, as it is biased by contemporary vector records. Authors should be careful of making statements about changes in vector favourability, or predictions of future favourability such as Lines 384-385 “models show both species expanded their favourable areas”. It is highly likely that these areas were always favourable for the presence of the species, however, the mosquito had either been eliminated from them previously (Aedes aegypti DDT interventions) or as in the case of Europe and Ae. albopictus, had been newly introduced. In both these cases the spread of populations is not related to changes in favourability in environmental characteristics and the authors need to acknowledge this limitation in their model framework.

Reviewer #2: See below

Reviewer #3: The objectives of the study are clearly stated, that is, to investigate the fine-scale geographic changes in transmission risk since the late 20th century, and take into account for the first time the potential contribution that primate biogeography and sylvatic vectors make to increasing the risk of the disease.

The study design and the population are appropriate. Statistical analysis used to support conclusions were correct. 

Although the methodology is appropriate, the categorization of favourability values need to be clarified.

**Results**

-Does the analysis presented match the analysis plan?

-Are the results clearly and completely presented?

-Are the figures (Tables, Images) of sufficient quality for clarity?

Reviewer #1: Line 365: S1 Fig A does not exist.

Line 415: Use of the phrase “favourable areas for dengue have spread” is a more appropriate way to use the favourable index derived in this manuscript. This is due to the necessity of the vector to be present for dengue to be transmitted. Authors should focus on this terminology when using changes in risk predicted by their models. However, it is quite hard to justify an increased risk of disease transmission in small areas like South Australia (Fig 3) which has never had any of the vectors or disease present, and not include large areas of south eastern United States (eg. Mississippi, northern Florida) which has had both Ae. aegypti and Ae. albopictus present historically and epidemics of dengue. The authors need to justify why these historical areas are considered favourable or not in their models, not draw attention to small areas of predicted “favourability”, and if unable to justify these, need to be described as a limitation. 

Line 418. Use of the work significant in scientific literature means that a statistical analysis was performed, especially in a results section. Please provide results of this analysis or remove this word to describe results.

Using the term “risk” is hard for readers to understand what risk actually means within the context on this manuscript unless it is clearly defined and used consistently through the manuscript for clarity. For instance, the Abstract mentions risk three times – transmission risk, risk of disease and risk factors. Authors needs to be consistent with this language throughout the model. 

Figure 3. The manuscript would benefit by ensuring all reference to the transmission model be referred to “dengue transmission risk model”. This will clear up to the reader what risk means in the context of this paper. This risk can then be referred to ask either increases in zoonotic and/or anthroponotic dengue transmission risk and the manuscript may benefit by highlighting these changes in different maps. 

Line 456: Assessment spelt incorrectly.

Line 458-459: Authors have used a value judgement here instead of stating the results directly. It should be up to the reader to judge the quality of the discrimination capacities as this sounds like confirmation bias.

Line 462: Why is risk used in this sentence? Does it relate to transmission risk? Please state this.

Lines 477-513: All commas in numbers need to be replaced with decimal places.

Line 488: remove the word “got”

Figure 4 is an informative map and should be improved. Replace commas in numbers. Use one legend on the figure. Light green areas look more yellow than light green. Legend for the light green areas on the figure is uninformative. The description used in the caption should be used instead.

Reviewer #2: See below

Reviewer #3: The results are clearly presented. However, some results need to be clarified or reorganized (the attached file is provide it). 

Table and some Figures need to be revised such as the map captions and figure captions.

**Conclusions**

-Are the conclusions supported by the data presented?

-Are the limitations of analysis clearly described?

-Do the authors discuss how these data can be helpful to advance our understanding of the topic under study?

-Is public health relevance addressed?

Reviewer #1: There are no conclusions. This is a flaw in the manuscript.

Reviewer #2: See below

Reviewer #3: The conclusion is stated but the limitations are not provided.

**Editorial and Data Presentation Modifications?**

Reviewer #1: Throughout the manuscript:

Aedes should be correctly abbreviated from A. to Ae.

Authors should consider the use of “Results predict” or “Our results suggest” instead of “We anticipate”.

Unless absolutely necessary to emphasise something, references are preferably used at the end of sentences.

What does quantitatively and qualitatively mean with respect to dengue cycles? The authors need to clearly define what they mean with this terminology. 

Abstract

Line 13: Remove “successfully”.

Lines 26: “distribution” not “distributions”.

Line 34: Suggest using “novel” instead of “new”.

Lines 29-30: Remove “the” from “in the dengue-transmission risk”.

Introduction

Combine paragraphs 2 and 3 into one paragraph for brevity.

Line 75: Transmitted spelt incorrectly. 

Line 89 and elsewhere. Re-emergence spelt incorrectly. 

Line 104: Remove “that”.

Lines 107-108: Remove “This is the objective of the present study”. This is evident.

Reviewer #2: See below

Reviewer #3: MAIN POINTS:

Lines 3-23: Based on submission guidelines (Manuscript organization),

“The Abstract is conceptually divided into the following three sections with these headings: Background, Methodology/Principal Findings, and Conclusions/Significance.”

I would suggest reorganizing the abstract following these guidelines.

Line 172: …” Variables with correlation coefficients >0.8 were not allowed…”.

How did you choose 0.8?

Lines 179-183, Fig 1: It is hard to distinguish between Bottom-left and Bottom-right. Please provide a more detailed explanation!

Lines 304-308: You stated that “ Favourability values were finally reinterpreted as transmission-risk values following this scale: high favourability (i.e. F ≥0.8 [69]) was referred as high transmission risk; favourability values between ≥0.5 and 0.8 were referred as intermediate-high risk; … and low favourability (i.e. F < 0.2 [69]) was referred as low transmission risk.

The reference that you refer to, that is, Numoz et all, (2005) mentioned that they classified favourability into only three categories as follows:

“If the predicted favourability was higher than 0.8, which means that the odds are more than 4:1 favourable to the species, the square was considered as favourable. Areas with a favourability value lower than 0.2 (odds less than 1:4) were considered unfavourable to the species. The remaining squares were considered as intermediate favourability areas.”

How did you categorize intermediate-high risk and intermediate-low risk? Please provide some more explanations!

If you follow this reference, 

line 305 should be … high favourability (i.e. F > 0.8 [69]) was referred as high transmission risk (without “equal” sign).

Line 306: “favourability values between ≥0.5 and 0.8” Should not be “0.5 � F < 0.8”?

Line 307: “between ≥0.2 and 0.5” Should not be “0.2 � F < 0.5”?

Line 460: CCR ranged between 0.771-0.884.

Do you mean “ CCR ranged between 0.771 and 0.874?

Lines 460-462: It is stated that “ Kappa values were >0.220 in all the 20th -century models, and >0.350 (never <0.500 in the disease and transmission models) in the 21st-century models.

What do you want to emphasize by including this sentence (never <0.500 in the disease and transmission models)…?

My suggestion is as follows:

Kappa values ranged between 0.228 and 0.252 in the 20th-century models and ranged between 0.352 and 0.533 in the 21st-century models.

Lines 467-468:…., the CCR values decreased by about 13%,…

How did you get 13%?

Line 498: The 21st century (2001-2017) models … record (Table 2).

However, in Table 2 it is stated 2000 to 2019. Which one is correct?

Line 515: A total of 51 chorotypes were detected: 24 chorotypes in Asia,…

Fig S4 is unclear. Asian chorotype names are coded as AS1 to AS24. However, we can only see clearly 12 codes among 24 code names. Please make it clearer!

Lines 516-519: “Two Asian chorotypes form part of the 20th-century disease model,….”

Is that correct that only two Asian chorotypes from part of the 20th-century disease model (AS8 and AS15)? What about AS7, AS9, and AS19? In S4 Fig you stated that “Violet rectangles: chorotypes characterizing the distribution of the 20th-century dengue cases….”

Lines 523-525: “…the 21st -century model also comprised, from Asia: Macaca in chorotype AS5; Hylobates, Presbytis, Symphalangus and Nycticebus in chorotype AS7, Loris, Semnopithecus and Macaca in chorotype AS9, and Carlito in chorotype AS19;”. Is that correct? Should not be only AS5 included?

Lines 525-527: “from Africa: … in chorotype AF2”. Should not be included in the 20th century?

Lines 514-528: Overall, I found the paragraph to be confusing. I would suggest that first explain the chrotypes in Asia (S4 Fig), then chrotypes in Africa (S5 Fig) and America (S6 Fig) of the 20th-century disease model followed by the 21st-century disease model. Please reorganized!

Lines 529-530: “Primate chorotypes contributed to explain a maximum of 16.4% of the variation in favourability for the presence of dengue in the 20th -century (Fig. 4)”. 

I would suggest change (Fig.4) to be (Fig. 4A) as this sentence only related to Fig. 4A

Line 534: “In the early 21st-century model, chorotypes contributed a maximum of 9.17% to explain the…”

Do you mean 9.8%? 0.7+9.1 =9.8

Lines 534-535: “the early 21st-century model, chorotypes contributed a maximum of 9.17% to explain the variation in favourability, although only a 0.7% could be exclusively attributed to them”.

I would suggest adding (Fig. 4B) to the end of this sentence.

Lines 543-544, Fig.4.: The map captions are too small for all 2 maps. Please make it readable! 

I would suggest making the maps bigger.

Lines 551-556: I think Figure captions are too long. I would suggest providing a more detailed explanation regarding Fig. 5 in a paragraph.

 

MINOR POINTS:

Line 164: …according to Rao’s score tests, …

I would suggest including a reference

Line 169: … and Wald tests

I would suggest including a reference

Lines 187, 236: …. georreferenced…

Should not be “georeferenced”?

Line 243: Spell out ECDC when it was first mentioned.

European Centre for Disease Prevention and Control (ECDC)

Line 247: Spell out PAHO

Line 248: Spell out GIDEON

Line 264: Spell out IUCN

Line 323: Bombi &d’Amen [70] shoul be Bombi & D’Amen [70]

Lines 330-331: “…area under the receiver-operating-characteristic curve (AUC) [71].”

Should not be “…area under the curve receiver-operating-characteristics (AUC- ROC) Curve [71]?

Line 638: ….whereas Messina et al. ….”

I would suggest including a reference

Line 647: …. by Messina et al. …”. 

I would suggest including a reference

Line 697: ….. zoonotic transmission risk (Fig. 4b).

It should be … (Fig.4B).

**Summary and General Comments**

Reviewer #1: The manuscript “Worldwide dynamic biogeography of zoonotic and anthroponotic dengue” builds several models to predict both the potential distribution of dengue vectors and the risk of dengue transmission across a number of years in the 20th and 21st century. Incorporating sylvatic cycles and a zoonotic component is an important issue and I believe this type of analysis will be important in predicting future emerging infectious disease hotspots. The authors have gone put considerable effort into incorporating these factors into their predictions, especially as sylvatic cycles are related almost entirely on environmental conditions. The manuscript would benefit from an increased focus on these challenges.

The manuscript has merit for publication in PLOS NTD, however, there are a number of issues that will need to be addressed before I believe it will reach the standard required. The manuscript is over 12,000 words and there are many areas where this can be reduced for brevity and clarity. For instance, English may not be the authors first language but many of the statements made throughout the manuscript need to be clarified, removed or simplified as they can be irrelevant to the narrative. Some of the priority issues are listed below.

A major problem with the manuscript is a failure to frame their results correctly in the context of Aedes historical distributions with respect to “favourability” and in particular with Aedes aegypti historical distributions. There has been extensive documentation on the distribution of this vector for over 100 years now, and yet the results in this manuscript are biased by the distribution they have chosen (from Kraemer et al 2015) which does not include records of the species at a time before DDT was used to eliminate large populations of this species. This includes areas in North America, South America, Australia and Europe which were clearly “favourable” to the species but not included in this model. As such authors first need to reframe their argument throughout the manuscript in terms of “late 20th century” and “early 21st century and not “20th century” or “21st century”. I noticed this has occurred in some places, including on figures but it is important for consistency that this terminology is used throughout the manuscript. This will have an impact on the way results and discussion are framed.

Linked to this issue is the confidence in which authors state how accurate their results are. No model is perfect, and modellers should be careful when making absolute statements or value judgements about the world or the quality of their results. By not including species distribution records from before 1970 the “favourability” model is already inaccurate as Ae. aegypti distributions were considerably larger before this time, confounding author’s conclusions. As Ae. aegypti is the primary vector of dengue, this considerably increased the risk of dengue transmission in the early 20th Century. For example, there were huge epidemics of dengue in Greece (~30,000 cases) and Australia (~100,000 cases) in the 1920/30s which impacted the majority of the population in affected areas. Aedes aegypti distribution is not linked entirely to environmental variables, but also artificial water storage by humans. It was artificial water storage which led to these large outbreaks in the early 20th century in areas of Australia and Greece where rainfall is not high enough for the species to survive on rainfall alone. This is not adequately represented in the model and must be stated as a limitation to conclusions. Furthermore, these areas in Australia are not currently represented as dengue transmission risk in Figures 2, and only barely become represented in Figure 3 and I believe this is a failure of the model.

Reviewer #2: Authors present a worldwide model of dengue transmission risk using updated vector and dengue case data. They included data from the sylvatic vectors of dengue and also chorotypes for non-human primates. The entire modeling is based on fuzzy logic (fuzzy sets) and logistic regressions with stepwise forward-backward model selection, for example, that is how they develop their chorotypes. The study is well written, well referenced and novel. However, they are evaluating the models based on discriminatory statistics without true absence data. A crucial problem on this type of analysis that should be explicitly declared, presented and dealt with, which is currently not the case. See further comments: 

Lines 329-336: Authors should provide a better justification of using AUC and the rest of discriminatory statistics based on presence/absence data. Even their own reference (i.e., 71) present important arguments on why those statistics should not be used for model selection. Specifically, absence data is lacking and the use of the entire globe as study area is for sure inflating their statistics, thus, authors should be more careful on the interpretation of their ‘excellent’ predictive power (line 596). This is a very important problem of the entire modeling framework since render their metrics useless. See also: Peterson et al 2008 (doi: 10.1016/j.ecolmodel.2007.11.008) and Jiménez et al 2020 (doi: 10.1111/2041-210X.13479). I will recommend to use instead omission rates (i.e. a performance metric; Galante et al 2017 (doi: 10.1111/ecog.02909), Cobos et al 2019 (doi: 10.7717/peerj.6281) and include in the discussion a section commenting on the difficulties of model selection on species distribution models (Velasco et al 2018 (doi: 10.1016/j.ecoinf.2019.02.005)). Using study areas specific for each vector might also be very useful (Barve et al 2011 doi: 10.1016/j.ecolmodel.2011.02.011). 

Authors should mention the number of points used for each model. In this line, please mention the number of occurrences available for each vector including the non-urban mosquito points. Also, please mention the total number of dengue cases used from Messina et al and how many more occurrences where added after your updated search. This information should be presented to have a clearer idea of the evaluation phase, for example, how many points where used for evaluating the 21st century model with 2018-2019 cases? 

It is not clear how did you include information from public health reports that are usually presented in the form of uninformative geographical centroids in big administrative units (e.g., provinces), instead of actual ‘points’; this might be problematic while transferring information to hexagons by adding inaccurate records to different regions. 

Line 123: Please specify that the first number corresponds to grid cells and the second number to the area represented by the hexagons. It makes sense that this approach avoids autocorrelation, however it is not clear if the post-calibration downscaling might introduce another source of uncertainty in your interpretations. 

Line 263: Eliminate word ‘primates’

Lines 199-202: Please make it clear which variables are used for which model in the supplementary material, so readers can refer only to the table to recognize variable application in just one section of the manuscript. This is a comment that should be done considering the refinement of the model (lines 310-318), because according to the main text, for example, livestock density was used for the vector but not for the disease models. 

Lines 317-318: Was the inclusion of the new set of variables a source of multicollinearity on your models? Were they assessed for collinearity with the other variables? 

Rectangle colors in Supplementary figure number five and six are red instead of violet as mentioned in the legend, please correct accordingly. Names of non-human primate species in Supplementary figure 6 are almost unreadable. Please make sure the figures have enough resolution. 

Lines 304-308: Please mention that these thresholds are arbitrarily selected and not possess any empirical evidence to reflect either transmission risk category. It is useful for interpretation and should be kept, just be explicit in the way it is working. 

Line 635: Please add the name of the algorithm used in this manuscript between parentheses. 

I would like to suggest building a figure similar to Figure 5 using the models of the 21st-century. It can be presented in the same figure 5 or as a supplementary material.

Reviewer #3: The paper describes the potential contribution of primate biogeography and of sylvatic vectors in increasing the risk of dengue. This research aims to build a map that quantified the current level of dengue transmission risk worldwide by including a zoonotic component as well as temporal stratification in the production of a dengue risk map.

The manuscript is generally well-written. However, some points need to be improved and clarified as mentioned above.

PLOS authors have the option to publish the peer review history of their article (what does this mean?). If published, this will include your full peer review and any attached files.

Reviewer #1: No

Reviewer #2: No

Reviewer #3: No
---

## [Decision Letter · Decision Letter 1]

7 May 2021

Dear Ms Aliaga,

Thank you very much for submitting your manuscript "Worldwide dynamic biogeography of zoonotic and anthroponotic dengue" for consideration at PLOS Neglected Tropical Diseases. As with all papers reviewed by the journal, your manuscript was reviewed by members of the editorial board and by several independent reviewers. The reviewers appreciated the attention to an important topic. Based on the reviews, we are likely to accept this manuscript for publication, providing that you modify the manuscript according to the review recommendations. 

The reviewers appreciate the effort put forth to improve this submission. However, there are some issues that remain to be addressed including a number of grammatical errors. Please remember that PLoS journals do not proof submissions so authors are responsible for ensuring proper language use. It might me helpful to have a native English speaker proof your manuscript before resubmitting.

Sincerely,

Michael R Holbrook, PhD

Associate Editor

Elizabeth Carlton

Deputy Editor

The reviewers appreciate the effort put forth to improve this submission. However, there are some issues that remain to be addressed including a number of grammatical errors. Please remember that PLoS journals do not proof submissions so authors are responsible for ensuring proper language use. It might me helpful to have a native English speaker proof your manuscript before resubmitting.

Reviewer's Responses to Questions

**Key Review Criteria Required for Acceptance?**

**Methods**

-Are the objectives of the study clearly articulated with a clear testable hypothesis stated?

-Is the study design appropriate to address the stated objectives?

-Is the population clearly described and appropriate for the hypothesis being tested?

-Is the sample size sufficient to ensure adequate power to address the hypothesis being tested?

-Were correct statistical analysis used to support conclusions?

-Are there concerns about ethical or regulatory requirements being met?

Reviewer #1: (No Response)

Reviewer #2: see below

Reviewer #3: Yes, the authors have carefully revised the manuscript.

**Results**

-Does the analysis presented match the analysis plan?

-Are the results clearly and completely presented?

-Are the figures (Tables, Images) of sufficient quality for clarity?

Reviewer #1: (No Response)

Reviewer #2: see below

Reviewer #3: Yes, the authors have carefully revised the manuscript

**Conclusions**

-Are the conclusions supported by the data presented?

-Are the limitations of analysis clearly described?

-Do the authors discuss how these data can be helpful to advance our understanding of the topic under study?

-Is public health relevance addressed?

Reviewer #1: The discussion would benefit greatly from a conclusion paragraph. A conclusion paragraph is your chance to have the last word on your findings. Conclusion paragraphs allow you to have the final say on the issues you have raised in your paper, to synthesize your thoughts, to demonstrate the importance of your ideas, and to propel your reader to a new view of the subject. It is also your opportunity to make a good final impression and to end on a positive note.

Reviewer #2: see below

Reviewer #3: Yes, conclusion is clearly stated and the authors have carefully added the limitations in the new version

**Editorial and Data Presentation Modifications?**

Reviewer #1: Multiple lines: Replace “transmissions” with “transmission” on lines 6, 77, 109, 114, 767, 769, 787, 799, 810.

Line 8: Remove “therefore”

Line 16: Incorrect spelling of Papua New Guinea.

Line 129: Replace “transmitter” with “vector”

Line 131: Replace “resistances” with “resistance”

Line 133: Sizeable spelt incorrectly

Line 212: Remove the second “the” from the reference title: “The global compendium of the

Aedes aegypti and Ae. albopictus occurrence”

Line 230: Replace “infrastructures” with “infrastructure”

Lines 254-256: Remove second “in the late 20th century” in the sentence

Line 359: Replace “human victims” with “human hosts”

Line 389: “non-human primates” spelt incorrectly

Line 443: Use “Aedes” instead of abbreviation “Ae” when starting a sentence

Line 618: Remove “by that time” from the sentence

Line 620: Remove “broadly”

Line 626: Define the type of risk you are referring to here

Line 628-659: Place the “limitations” paragraph later in the discussion. You want to start the discussion with your most important results.

Line 633: Place “transmission” in front of risk and remove from after dengue.

Line 660: Remove the first phrase about methodological limitations. It is no longer needed here

673: Incorrect spelling of albopictus.

Line 729: Change “South” to “southern”

Line 732: Remove “alone”

Line 756: Informal language “sound alarm bells louder”. Change to “sound a warning”

Line 757-758: Remove hyphens and use commas

Line 766: Remove “the”

Line 772: Remove “being”

Line 797: Rephrase “primate chorotypes was pure” to “pure primate chorotypes”

Line 799: Replace “mapped” with “estimated”

Line 820-823: Rewrite sentence to “This concentration of evidence suggest the need for studies that address the occurrence of dengue sylvatic cycles in the Atlantic forest of Brazil and the Amazon”

Reviewer #2: see below

Reviewer #3: (No Response)

**Summary and General Comments**

Reviewer #1: The authors have done a good job in responding to reviewer comments and the paper is almost finished. There is still a number of spelling mistakes, and poor English in the manuscript which will need to be addressed before it can move to publication. The incorrect use of plurals still needs attention and the discussion would benefit from a concluding paragraph.

Reviewer #2: Authors have argued favorably to all my comments and I believe the manuscript is now suitable for publication. Regardless, I will insist to include a comment in the point referring to the discriminatory capacity of their models using AUC: 

In the paper of Lobo et al., they clearly argue the following: “increasing the geographical extent outside presence environmental domain entails obtaining higher AUC scores” which is clearly something that is happening while using the ‘entire world’ as a calibration area for your modeling effort. Thus, your now ‘outstanding’ discrimination capacity, lies on the inflation of statistics due to including a larger calibration area. The implementation of a ‘trend-surface’ does not fix this problem since you are including it as another co-variate during you model calibration step which is using the entire world for the rest of the predictors as well. Your other statistics based on thresholds (currently justified) are a fair representation of the performance of the models and the majority of your interpretations are based on those results. I will like to encourage you to be explicit on the problems of the AUC to actually invite other authors to avoid using it in the context of lack of true absence data and the inflation of his values due to using calibration areas beyond the accesible area of the studied species (Jiménez & Soberón, 2020: https://doi.org/10.1111/2041-210X.13479)

Check typos and old information in Supplementary Table 3: The word should be ADMINISTRATIVE. Also, reference to Supplementary figures in the Primate chorotype section is outdated, it should be Supp. Figs 5-7. 

Lines 418-423. Please consider highlighting the variables mentioned in the main text in the Supplementary Tables S4, S5, and S6. 

Lines 440-449: Please consider including a table showing the contribution of your predictors to the favorability models of the dengue sylvatic vectors (as is done with Supp Table 4, 5, or 6). 

Line 534: Favorability instead of favorabity

Line 615: It is not clear what the authors meant with ‘fine-scale’ 

Line 625: The authors are not suggesting any specific management strategies, they are showing potential risk factors. 

 Line 654: Should be ‘still’

Line 659: Refer the reader to the appropriate figures and supplementary material. 

Make sure that Fig S8 is mentioned in the main text.

Reviewer #3: (No Response)

PLOS authors have the option to publish the peer review history of their article (what does this mean?). If published, this will include your full peer review and any attached files.

Reviewer #1: No

Reviewer #2: No

Reviewer #3: No

Figure Files:

Data Requirements:

Reproducibility:

References

---

## [Editor Report · Decision Letter 2]

22 May 2021

Dear Ms Aliaga,

We are pleased to inform you that your manuscript 'Worldwide dynamic biogeography of zoonotic and anthroponotic dengue' has been provisionally accepted for publication in PLOS Neglected Tropical Diseases.

Best regards,

Michael R Holbrook, PhD

Associate Editor

Elizabeth Carlton

Deputy Editor

---

## [Editor Report · Acceptance letter]

3 Jun 2021

Dear Ms Aliaga-Samanez,

We are delighted to inform you that your manuscript, "Worldwide dynamic biogeography of zoonotic and anthroponotic dengue," has been formally accepted for publication in PLOS Neglected Tropical Diseases.

Best regards,

Shaden Kamhawi

co-Editor-in-Chief

Paul Brindley

co-Editor-in-Chief
